# Graph is a Natural Regularization: Revisiting Vector Quantization for Graph Representation Learning

**Zian Zhai** [1]  **Fan Li** [1]  **Xingyu Tan** [1]  **Xiaoyang Wang** [1]  **Wenjie Zhang** [1]

## Abstract

Vector Quantization (VQ) has recently emerged as a promising approach for learning compressed and discrete representations for graph-structured data. However, a fundamental challenge, i.e., codebook collapse, remains underexplored in the graph domain, significantly limiting the expressiveness and generalization of graph tokens. In this paper, we present an empirical study and observe that codebook collapse consistently occurs when training VQ jointly with Graph Neural Networks under graph reconstruction tasks, even with mitigation strategies proposed in vision or language domains. Moreover, we provide a diagnosis of collapse from data and optimization perspectives, showing that collapse is associated with graph data properties such as feature redundancy and connectivity density, and is further reinforced by the training dynamics of deterministic hard assignment. To address these issues, we propose RGVQ, a novel framework that integrates graph topology and feature similarity as explicit regularization signals to enhance codebook utilization and promote token diversity. RGVQ introduces soft assignments via Gumbel-Softmax reparameterization, ensuring that all codewords receive gradient updates. In addition, RGVQ incorporates a structure-aware contrastive regularization to penalize assigning the same token to dissimilar node pairs. Extensive experiments demonstrate that RGVQ substantially improves codebook utilization and consistently boosts the performance of state-of-the-art graph VQ backbones across multiple downstream tasks, enabling more expressive and transferable graph token representations.

## 1. Introduction

In recent years, a discretization-based tokenization method, known as Vector Quantization (VQ), has attracted significant research attention for its effectiveness in generative modeling (Van Den Oord et al., 2017; Caron et al., 2018). VQ quantizes continuous latent representations into discrete clusters referred to as "codewords" in a learnable codebook (Zhang et al., 2023). These codewords are then trained to reconstruct the original data samples. By discretizing latent space, VQ provides an effective prior for learning disentangled features, and has achieved remarkable success in generative tasks, including image synthesis (Ramesh et al., 2021; Chang et al., 2023; Li et al., 2024a), speech generation (Dhariwal et al., 2020; Zhang et al., 2024), and language models (Liu et al., 2025; Van Baalen et al., 2024).

Motivated by these successes, recent efforts have begun to explore the extension of VQ to graphs for scalable and versatile graph tokenization. First, discretizing graphs into VQ tokens enables compact graph compression, substantially reducing the memory and computation overhead during inference (Yang et al., 2024; Luo et al., 2025). Second, VQ provides a natural mechanism for abstracting structural patterns into a reusable token vocabulary, analogous to the language tokens used in Large Language Models (LLMs), and offers a promising pathway toward Graph Foundation Models (GFMs) (Wang et al., 2024). Third, VQ allows graphs to be serialized into token sequences, enabling sequence-based modeling with standard Transformer architectures that are widely adopted in NLP and vision, and eliminating the need for handcrafted inductive biases that are typically required in Graph Transformers (Wang et al., 2025).

Similar to VQ models in the vision and language domains, which are typically trained to reconstruct input samples jointly with encoders (Navaneet et al., 2024; Deng et al., 2025), Graph VQ is likewise trained jointly trained with Graph Neural Networks (GNNs) under graph reconstruction objectives, including both node feature and edge reconstruction. Nevertheless, through our empirical study, we observe that **codebook collapse** consistently occurs, even when applying mitigation strategies that are commonly used in other domains. This refers to the phenomenon where most inputs are mapped to only a few codewords, leaving the majority

---

[1]School of Computer Science and Engineering, University of New South Wales, Sydney, Australia. Correspondence to: Fan Li <fan.li8@unsw.edu.au>.

underutilized (Zhu et al., 2025; Zhang et al., 2023). As a result, only a limited number of tokens can be utilized during inference, leading to overly coarse representations and degradation in task performance. However, prior work focuses only on the performance of downstream tasks, without addressing this critical problem in the first place (Wang et al., 2024; Zeng et al., 2025; Yang et al., 2024). This consistent underutilization naturally raises a central question: *What makes Graph VQ more prone to collapse?*

To answer this question, we diagnose the collapse from both *data* and *optimization* perspectives. From the data perspective, we find that the severity of collapse is associated with typical graph properties such as feature redundancy and connectivity density. In particular, graphs with higher feature redundancy and denser connectivity are linked to more severe collapse, suggesting that intrinsic properties of graph data can exacerbate the issue. From the optimization perspective, we analyze the training dynamics of deterministic VQ and show that hard assignment induces a self-reinforcing feedback loop: frequently selected codewords receive more updates and become increasingly dominant, while rarely selected ones remain inactive, which limits utilization exploration and ultimately drives the system toward collapse.

Based on these insights, we propose **Regularized Graph Vector Quantization** (**RGVQ**), a novel framework that integrates graph topology and feature similarity as explicit regularization signals to enhance codebook utilization. First, to break the self-reinforcing loops, RGVQ adopts the Gumbel-Softmax reparameterization to relax hard assignments into differentiable probability distributions, enabling the gradients to flow not only to the most likely codewords but also to the less probable candidates. Second, RGVQ leverages graph topology and feature similarity to regularize token assignment distributions, explicitly penalizing overly concentrated token utilization induced by graph redundancy. This regularization encourages nodes with similar features and local structures to share token distributions, while discouraging similar assignment among unrelated nodes.

Our contributions can be summarized as follows.

- To the best of our knowledge, we provide a systematic empirical study of codebook collapse on graphs, benchmarking mitigation strategies from other domains and revealing collapse as a fundamental bottleneck in discrete graph token learning.

- We provide a collapse diagnosis from data and optimization perspectives, and identify that collapse is associated with graph redundancy and hard-assignment in deterministic VQs.

- We propose RGVQ to address graph data redundancy by structure-aware regularization and disrupts the self-reinforcing dynamics via stochastic quantization.

- We perform comprehensive experiments on state-of-the-art (SOTA) Graph VQ backbones, demonstrating that our proposed method improves codebook utilization and downstream performance, and serves as a flexible plug-in for learning graph tokens.

## 2. Related Work

**Vector Quantization**. Vector Quantization (VQ) maps continuous inputs to discrete tokens in a codebook and has been widely used in image, video, and audio generation (Chung et al., 2020; Fifty et al., 2025; Tang et al., 2022). This success has motivated efforts to extend VQ to graph data. For example, VQ-GNN (Ding et al., 2021) and VQGraph (Yang et al., 2024) apply VQ for embedding compression, but their fully supervised training deviates from the original unsupervised training scheme of VQ (Chen & Lee, 2021; Yu et al., 2022). More recently, GFT pretrains VQ by reconstructing graph features to utilize the learned codebook as transferable vocabulary across tasks and domains (Wang et al., 2024). GQT employs residual VQ to tokenize graphs for vanilla transformers, alleviating manual architectural bias in graph transformers (Wang et al., 2025). While both methods demonstrate promising applications of Graph VQ, they overlook the issue of codebook collapse, which undermines the generalization of learned tokens. HQA-GAE introduces a hierarchical VQ, improving performance on graph tasks (Zeng et al., 2025). Nevertheless, it does not effectively resolve the non-differentiability of VQ and lacks a formal analysis of codebook collapse.

**Collapse Mitigation**. One of the most fundamental limitations of VQ is codebook collapse, wherein only a small fraction of codewords are used (Zhang et al., 2024; Lu et al., 2023). Various mitigation strategies have been explored. Exponential Moving Average (EMA) is proposed to stabilize codebook updates (Polyak & Juditsky, 1992; Wu & Yu, 2019). Pretraining the encoder (Zhao et al., 2024) is proposed to mitigate embedding drift during training VQ. In addition, codebook reset (Zeghidour et al., 2021; Williams et al., 2020) periodically reinitializes inactive codewords with encoder embeddings. Affine parameters (Huh et al., 2023; Zhang et al., 1997) introduce a learnable transformation to align encoder outputs with the codebook space. Recently, SimVQ (Zhu et al., 2025) reparameterizes the code vectors through a linear transformation layer based on a learnable latent basis. Although these mitigation strategies have been evaluated in image and speech domains, their performance on graph data remains underexplored.

## 3. Preliminary

**Graph Neural Network**. Graph Neural Networks (GNNs) learn the node representations by recursively aggregat-

ing features from neighbors, also known as message-passing (Zhai et al., 2025; Tan et al., 2026a;b; Li et al., 2026b). Formally, the representation of node $v$ at the $l$-th layer is:

$$\mathbf{h}_v^{(l)} = \text{AGG}(\{\mathbf{h}_u^{(l-1)}, u \in \mathcal{N}(v) \cup v\}, \phi^{(l)}), \quad (1)$$

where $\mathbf{h}_v^{(0)} = \mathbf{x}_v$ is the initial node feature, $\mathcal{N}(v)$ is the neighbor set of node $v$, and $\phi^{(l)}$ is the parameters of the $l$-th layer of the GNN. The aggregation function $\text{AGG}(\cdot)$ combines the embedding of node $v$ and its neighbors, which is typically implemented as sum, mean, or max pooling (Xu et al., 2026; Li et al., 2026a; Zheng et al., 2025).

**Deterministic VQ**. VQ maps continuous vectors into a finite set of discrete embeddings in the codebook (Van Den Oord et al., 2017). Given a codebook $\mathbf{C} = [\mathbf{e}_1, ..., \mathbf{e}_K] \in \mathbb{R}^{K \times d}$ with each discrete codeword $\mathbf{e}_i \in \mathbb{R}^d$, a continuous input $\mathbf{h}_i \in \mathbb{R}^d$ is quantized as $\mathbf{z}_i$ with the nearest codeword $\mathbf{e}_k$ by:

$$k = \arg\min_j \|\mathbf{h}_i - \mathbf{e}_j\|_2^2 = \arg\min_j \|\mathbf{h}_i - \delta_j \mathbf{C}\|_2^2, \quad (2)$$

where $\delta_j \in \{0, 1\}^{1 \times K}$ is the one-hot indicator vector with only the $j$-th element being 1. To enable gradient propagation through the non-differentiable vector $\delta_j$, the Straight-Through Estimator (STE) is applied (Bengio et al., 2013). During the backward process, the gradient of the quantized embedding $\mathbf{z}_i = \delta_j \mathbf{C}$ is copied to $\mathbf{h}_i$, which is denoted as

$$\mathbf{z}_i = \text{sg}[\delta_j \mathbf{C} - \mathbf{h}_i] + \mathbf{h}_i, \quad \Rightarrow \frac{\partial \mathbf{z}_i}{\partial \mathbf{h}_i} = 1, \quad (3)$$

where $\text{sg}[\cdot]$ denotes the stop-gradient operator, ensuring the gradient for one-hot selection $\delta_j \mathbf{C}$ is discarded during the backward process. Finally, the learning objective is to reconstruct the input samples, with a codebook loss that pulls the quantized representations $\mathbf{Z} = [\mathbf{z}_1, \mathbf{z}_2, \ldots, \mathbf{z}_N]$ toward the encoder outputs $\mathbf{H} = [\mathbf{h}_1, \mathbf{h}_2, \ldots, \mathbf{h}_N]$, and a commitment loss that pulls the encoder outputs toward the quantized representations:

$$\mathcal{L}_{\text{VQ}} = \underbrace{\mathcal{L}_{\text{recon}}}_{\text{reconstruction loss}} + \underbrace{\|\text{sg}[\mathbf{H}] - \mathbf{Z}\|_2^2}_{\text{codebook loss}} + \beta \underbrace{\|\mathbf{H} - \text{sg}[\mathbf{Z}]\|_2^2}_{\text{commitment loss}}. \quad (4)$$

For Graph VQ, the reconstruction task typically involves reconstructing the graph properties, i.e., node features and links (Wang et al., 2024; 2025; Yang et al., 2024):

$$\mathcal{L}_{\text{recon}} = \underbrace{\frac{1}{N} \left\| \mathbf{X} - \hat{\mathbf{X}} \right\|_2^2}_{\text{feature reconstruction}} + \underbrace{\left\| \mathbf{A} - \hat{\mathbf{A}} \right\|_2^2}_{\text{link reconstruction}}, \quad (5)$$

where $\mathbf{A} \in \mathbb{R}^{N \times N}$ denotes the adjacency matrix, $N$ is the total number of nodes, and $\mathbf{X} \in \mathbb{R}^{N \times D}$ is the node feature matrix. The reconstructed feature matrix $\hat{\mathbf{X}} = g_{\theta_1}(\mathbf{Z})$ and

the reconstructed adjacency matrix $\hat{\mathbf{A}} = g_{\theta_2}(\mathbf{Z})$ are generated by the decoders for the feature reconstruction and link reconstruction tasks, respectively (Wang et al., 2024; Zeng et al., 2025).

**Metric for Codebook Utilization**. The extent of codebook collapse is measured by the codebook perplexity (Takida et al., 2022; Yan et al., 2024; Zheng & Vedaldi, 2023), which is defined as:

$$P = \exp\left(-\sum_{k=1}^K p_k \log p_k\right), \quad (6)$$

where $p_k$ denotes the probability of selecting the $k$-th codeword. A low perplexity indicates that only a few codewords dominate the assignments, reflecting a high degree of collapse. In contrast, a high perplexity suggests better utilization of the codebook capacity.

## 4. Motivation

Recent studies demonstrate the potential of Graph VQ; however, most existing methods primarily focus on downstream performance, leaving codebook utilization largely underexplored. This limitation becomes a bottleneck that hinders the scalability of graph tokens with different codebook size, thereby preventing flexible choices between expressive and compressed tokenization. Moreover, overly concentrated token assignments also impair downstream performance, as shown in Section 6.2. In this section, we conduct an empirical study of the codebook utilization and observe that codebook collapse occurs consistently during training, even when applying SOTA mitigation strategies from the language and vision domains. Furthermore, We diagnose this phenomenon from both data and optimization perspectives. From a data perspective, we find that the severity of collapse correlates with the properties of graph data: higher feature redundancy and denser local connectivity are associated with lower codebook utilization. From an optimization perspective, we show that deterministic hard assignment induces a self-reinforcing feedback loop, under which rarely selected codewords receive little update and become difficult to reactivate once they fall behind, thereby limiting effective exploration of the codebook. These insights motivate the development of our method.

### 4.1. Empirical Study

We begin by investigating the codebook perplexity of Graph VQ on different graph datasets. Following the settings of prior work (Ding et al., 2021; Wang et al., 2024; Yang et al., 2024), we apply vanilla Graph VQ and its variants augmented with SOTA collapse mitigation methods, including EMA, codebook reset, pretrained encoder, and affine parameters. By default, orthogonal normalization (Yu et al., 2022)

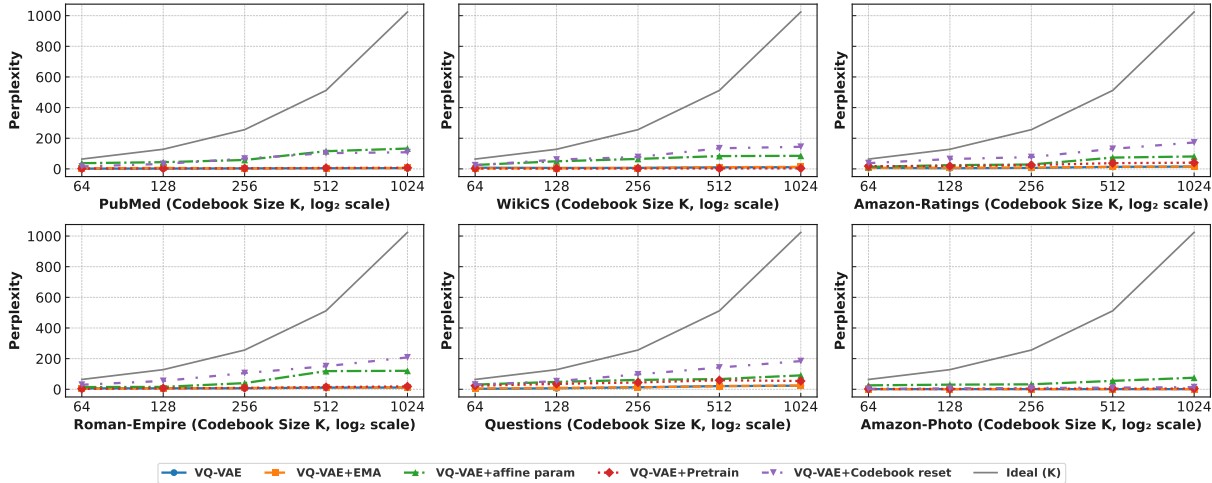

*Figure 1.* Codebook perplexites on graph datasets. The black lines indicate the optimal perplexities, i.e., codebook size K.

and cosine similarities (Wang et al., 2024) are incorporated in all variants. The implementation details and additional datasets can be found in Appendix A and B, respectively. From Figure 1, we make the following observations: **(Ob. 1)** Codebook collapse is a systematic and severe issue in Graph VQ. Across all datasets, the perplexity of VQ remains far below the codebook capacity, and fails to grow proportionally with the increasing codebook size. **(Ob. 2)** General mitigation strategies adopted from other domains only achieve marginal improvements and fail to fundamentally address codebook collapse in graphs. These findings reveal that codebook collapse is not merely incidental, but a systematic issue in Graph VQ.

### 4.2. Collapse Diagnosis

**Graph Properties**. As codebook collapse consistently occurs in graphs, we hypothesize that the unique properties of graph data, i.e., inherent feature redundancy and non-i.i.d. nature, may contribute to this phenomenon. To investigate this, we analyze two graph-level statistics that serve as proxies for these properties on investigated datasets. We consider: (1) PCA@95%, which quantifies feature redundancy by measuring the number of principal components needed to preserve 95% of node feature variance (Dong et al., 2022; Hou et al., 2023); and (2) average node degree, which reflects local connectivity density. Higher degrees imply stronger dependencies between neighboring nodes, violating the i.i.d. assumption and serving as a simple proxy for the non-i.i.d. nature of graph data (Yang et al., 2024; Wu et al., 2019). From Figure 2, we observe a positive correlation between PCA@95% and codebook perplexity, and a negative correlation between average degree and perplexity. Lower PCA@95% suggests higher feature redundancy, and higher average degree implies stronger local connectivity and non-i.i.d. characteristics, both associated with a

greater tendency toward collapse. Additionally, we provide dataset statistics (PCA@95, average degree) and measured codebook perplexity across 8 graph datasets in Appendix B.

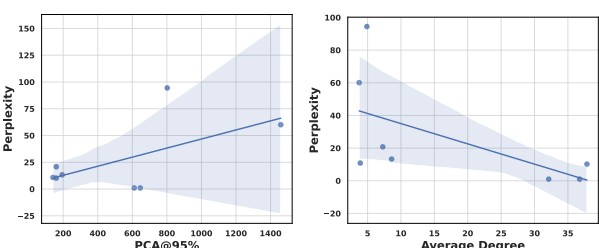

*(a)* PCA@95% vs. Perplexity   *(b)* Avg. Degree vs. Perplexity

*Figure 2.* Correlation between graph properties and perplexity.

**Optimization Dynamics**. To complement the data-side diagnosis, we examine the optimization dynamics of deterministic VQ and show how hard assignment can create a positive feedback loop that drives codebook collapse. In Graph VQ, the codebook $\mathbf{C}$ is updated only through the vocabulary loss (Zhu et al., 2025), i.e., the second term in Equation 4. The update is denoted as:

$$\mathbf{C}^{(t+1)} = \mathbf{C}^{(t)} - \eta \, \mathbb{E}_{\mathbf{h}_i} \left[ \delta_k^\top \delta_k \, \mathbf{C}^{(t)} \right] + \eta \, \mathbb{E}_{\mathbf{h}_i} \left[ \delta_k^\top \, \mathbf{h}_i \right], \quad (7)$$

where $\mathbf{h}_i$ is the embedding of node $v_i$, $\eta$ is the learning rate, and $\delta_k^\top \delta_k$ is the Kronecker delta matrix, defined as:

$$(\delta_k^\top \delta_k)_{ij} = \begin{cases} 1 & \text{if } i = j = k, \\ 0 & \text{otherwise.} \end{cases} \quad (8)$$

This condition indicates that if and only if when the expectation $\mathbb{E}_{\mathbf{h}_i}[\delta_k^\top \delta_k] = \mathbf{I}$, i.e., every token is selected with

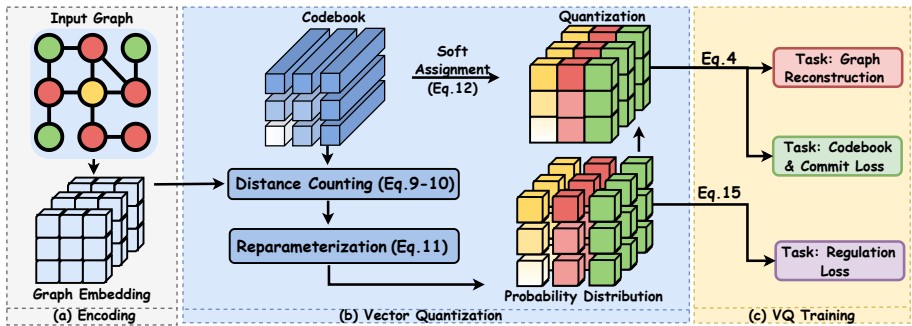

*Figure 3.* Overall framework of RGVQ. Note: red nodes represent the positive set, while green nodes denote negative samples.

equal probability $\frac{1}{K}$, each codebook entry is updated during training. In practice, according to the codebook loss term in Equation 4, selected codewords are updated and pulled towards the distribution of the output of the GNN, i.e., $\mathbf{h}_i$. On the other hand, the encoder outputs are simultaneously optimized towards the selected codewords via the commitment loss in Equation 4. This hard assignment and bidirectional attraction form a self-reinforcing "cocoon effect," which not only locks the encoder into preferring codewords, but also suppresses any possibility of unused codewords, specifically $(\mathbf{I} - \mathbb{E}_{\mathbf{h}_i}[\delta_k^\top \delta_k])\mathbf{C}$, being reactivated.

## 5. Methodology

Our collapse diagnosis identifies two key contributors to codebook collapse: (i) graph data properties and (ii) the self-reinforcing training dynamics of deterministic VQ with hard assignment. Accordingly, we propose RGVQ, a regularized Graph VQ framework consisting of two corresponding components, as illustrated in Figure 3. First, to break the optimization loop of deterministic VQ, RGVQ replaces hard assignments with differentiable assignment distributions among all codewords using Gumbel-Softmax reparameterization, enabling gradients to flow to all codewords proportionally to their assignment probabilities. Second, to mitigate the effect of graph data redundancy, RGVQ leverages graph topology and feature similarity to regularize token assignment distributions, explicitly discouraging overly concentrated token usage from dissimilar nodes and promoting higher token diversity. RGVQ can be added to the loss in Equation 4. Additionally, we provide the detailed sampling procedures and complexity in Appendix C.

**Gumbel-Softmax Reparameterization**. In deterministic VQ, the training dynamics of hard assignments prevent gradient backpropagation to unselected codewords, ultimately leaving them inactive and underutilized. To address this issue, we adopt Gumbel-softmax reparameterization (Roy et al., 2018; Sønderby et al., 2017), which replaces hard nearest-neighbor assignment with a differentiable soft selection. Formally, given node embedding $\mathbf{h}_i$, we define its

assignment logit $\boldsymbol{\pi}_i \in \mathbb{R}^K$ with each element computed by

$$\pi_{ik} = -\|\mathbf{h}_i - \mathbf{e}_k\|_2^2, \quad k = 1, \ldots, K, \qquad (9)$$

and the assignment probability vector $\mathbf{p}_i \in \mathbb{R}^K$ is denoted as:

$$\mathbf{p}_i = \text{Softmax}(\boldsymbol{\pi}_i). \qquad (10)$$

Instead of using a non-differentiable argmax over the distribution, we apply the Gumbel-Softmax trick to estimate a differentiable approximation of this hard assignment. Specifically, the assignment distribution is perturbed with Gumbel noise and passed through a temperature-controlled softmax:

$$\tilde{\mathbf{p}}_i = \text{Softmax}_\tau (\log \mathbf{p}_i + \mathbf{g}_i), \qquad (11)$$

where $\mathbf{g}_i \in \mathbb{R}^K$ is a noise vector with i.i.d. entries $g_{ik} \sim \text{Gumbel}(0, 1)$, and $\tau$ is the temperature. Given this estimated distribution, the quantized embedding is computed as a weighted average over all codebook entries:

$$\tilde{\mathbf{z}}_i = \mathbf{C}^\top \tilde{\mathbf{p}}_i. \qquad (12)$$

Because $\tilde{\mathbf{z}}_i$ is a differentiable weighted sum of all codewords, gradients propagate to each $\mathbf{e}_k$ proportionally to $\tilde{p}_{ik}$, rather than only to the selected entry in deterministic VQ, thereby mitigating the aforementioned self-reinforcing loop. During inference, the model reverts to deterministic hard assignment by selecting the codeword with the maximum logit $j = \arg\max_j \pi_{ij}$ and setting $\mathbf{z}_i = \mathbf{e}_j$.

**Structure-Aware Regularization**. To mitigate feature redundancy and dense connectivity in graph data, we incorporate feature and structural similarities to regularize the token assignment distribution computed from Gumbel-Softmax Reparameterization, encouraging the model to avoid overuse of some specific codebook entries. Our key insight is that collapse arises when nodes are spuriously mapped to the same tokens due to overly similar embeddings caused by dense connectivity or similar features. Therefore, we explicitly distinguish between similar and dissimilar node pairs based on both feature and local connectivity: similar nodes can exhibit more consistent assignment distributions, while

---

**Algorithm 1** Training procedure of RGVQ

---

**Input:** Encoder $f_\phi$, Decoder $g_\theta$, Codebook $\mathbf{C} = [\mathbf{e}_1, \ldots, \mathbf{e}_K]$, Temperature $\tau$, commitment weight $\beta$.
Initialize codebook $\mathbf{C}$ using K-means.
Compute positive set $\mathcal{N}_P$ and negative set $\mathcal{N}_N$ for each node.
**repeat**
    Sample minibatch $\mathbf{x} \sim p_{\text{data}}$.
    Compute embeddings $\mathbf{h} \leftarrow f_\phi(\mathbf{x})$.
    Compute logits $\pi_k \leftarrow -\|\mathbf{h} - \mathbf{e}_k\|^2$ for $k = 1, \ldots, K$.
    Assignment distribution $\mathbf{p} \leftarrow \text{Softmax}(\boldsymbol{\pi})$.
    Sample Gumbel noise $\mathbf{g} \in \mathbb{R}^K$ with i.i.d. $g_k \sim \text{Gumbel}(0, 1)$.
    Soft assignment $\tilde{\mathbf{p}} \leftarrow \text{Softmax}_\tau(\log \mathbf{p} + \mathbf{g})$.
    Soft quantization $\tilde{\mathbf{z}} \leftarrow \mathbf{C}^\top \tilde{\mathbf{p}}$.
    Update parameters by minimizing
        $\mathcal{L} = \mathcal{L}_{\text{recon}} + \mathcal{L}_{\text{reg}} + \|\text{sg}[\mathbf{H}] - \tilde{\mathbf{Z}}\|^2 + \beta\|\mathbf{H} - \text{sg}[\tilde{\mathbf{Z}}]\|^2$.
**until** Converge

---

dissimilar nodes should be discouraged from sharing similar distributions. Formally, given an anchor node $v$, we define:

- **Positive set $\mathcal{N}_P$**: consists of $n$ sampled nodes that are either structurally or semantically similar to $v$. Specifically, $n$ positive nodes are sampled from the union of the following two candidate sets: (1) nodes directly connected to $v$; or (2) the top $K$ feature-similar nodes to $v$. Formally, the positive set is denoted as:

$$\mathcal{N}_P = \{ u \mid (a_{uv} = 1) \vee \left( u \in \arg\text{topk}_{u' \in \mathcal{V}} \text{sim}(\mathbf{x}_{u'}, \mathbf{x}_v) \right) \}, \quad (13)$$

where $a_{uv} \in \mathbf{A}$ is the adjacency matrix, $\text{sim}(\cdot, \cdot)$ is the similarity function, $\mathbf{x}_v$ is the feature of node $v$. We apply the cosine similarity as the similarity function.

- **Negative set $\mathcal{N}_N$**: consists of $n$ sampled nodes that are neither structurally connected nor semantically similar to $v$. Formally, the negative set is defined as:

$$\mathcal{N}_N = \{ u \mid (a_{uv} = 0) \wedge \left( u \notin \arg\text{topk}_{u' \in \mathcal{V}} \text{sim}(\mathbf{x}_{u'}, \mathbf{x}_v) \right) \wedge (u \neq v) \}. \quad (14)$$

We encourage nodes in the positive set to have similar assignment distributions, while penalizing nodes in the negative set for having overlapping token distributions. Formally, given two nodes $v_i$ and $v_j$ and their token assignment distributions $\tilde{\mathbf{p}}_i$ and $\tilde{\mathbf{p}}_j$, the distributions are regularized by an InfoNCE loss (You et al., 2021; Wu et al., 2021), which is defined as:

$$\mathcal{L}_i = -\log \frac{\sum_{j \in \mathcal{N}_P} \exp(\text{sim}(\tilde{\mathbf{p}}_i, \tilde{\mathbf{p}}_j))}{\sum_{j \in \{\mathcal{N}_P \cup \mathcal{N}_N\}} \exp(\text{sim}(\tilde{\mathbf{p}}_i, \tilde{\mathbf{p}}_j))}. \quad (15)$$

We sum $\mathcal{L}_i$ all nodes to obtain the final regulation loss, i.e., $\mathcal{L}_{\text{reg}} = \sum_N \mathcal{L}_i$. The detailed training procedure can be found in Algorithm 1. The proposed regularization term is added to the reconstruction loss in Equation 4, forming the ultimate loss:

$$\mathcal{L}_{\text{VQ}} = \mathcal{L}_{\text{recon}} + \|\text{sg}[\mathbf{H}] - \tilde{\mathbf{Z}}\|^2 + \beta\|\mathbf{H} - \text{sg}[\tilde{\mathbf{Z}}]\|^2 + \mathcal{L}_{\text{reg}}. \quad (16)$$

## 6. Experiments

We evaluate the performance of RGVQ in terms (1) codebook utilization, (2) transferability, and (3) serialization. First, we evaluate the codebook utilization to verify its ability to mitigate codebook collapse. Second, we investigate the transferability. We integrate RGVQ into GFT (Wang et al., 2024), a graph foundation model that utilizes the learned codebook as pretrained graph tokens, and evaluate the performance on cross-task and cross-domain graphs. Third, we assess the serialization capability of RGVQ by examining its compatibility with sequence-based models. We use GQT (Wang et al., 2025), a transformer taking VQ tokens as input sequences, and evaluate the performance on node classification. Detailed dataset statistics, baselines, implementation details are provided in Appendix A. Experiments on large graphs can also be found in Appendix C.

### 6.1. Performance Evaluation

**Codebook Utilization**. We evaluate codebook utilization by comparing vanilla Graph VQ and its variants: EMA (Łańcucki et al., 2020), affine parameters (AP) (Huh et al., 2023), codebook reset (Reset) (Zeghidour et al., 2021), and pretrained encoders (PT) (Zhao et al., 2024), as well as existing SOTA VQ models: SimVQ (Zhu et al., 2025) and HQA-GAE (Zeng et al., 2025). All methods use orthogonal normalization, cosine similarity, and K-Means initialization (Im & Chan, 2023), and are trained on the feature and link reconstruction tasks. We fix the codebook size at 512

*Table 1.* Codebook utilization on homophilous and heterophilous graphs with codebook size $K = 512$. **Bold** highlights the best performance.

| | Cora | PubMed | Citeseer | Photo | Computer | WikiCS | Ratings | Roman | Questions |
|---|---|---|---|---|---|---|---|---|---|
| Graph VQ | 94.47±8.65 | 4.14±1.03 | 60.09±5.59 | 1.00±0.00 | 1.00±0.00 | 10.18±2.13 | 13.29±2.89 | 10.84±3.48 | 20.78±3.65 |
| EMA | 91.68±9.17 | 5.12±1.46 | 55.15±6.73 | 1.00±0.00 | 1.00±0.00 | 11.27±3.36 | 9.12±2.33 | 6.20±3.24 | 14.15±3.51 |
| AP | 75.32±6.28 | 126.55±12.64 | 9.03±2.16 | 54.95±5.43 | 59.33±8.55 | 83.55±9.86 | 73.82±8.14 | 118.46±16.21 | 66.57±8.27 |
| Reset | 65.79±8.56 | 102.78±15.78 | 85.19±4.41 | 10.73±1.98 | 17.18±2.37 | 134.44±7.35 | 130.83±8.88 | 150.51±11.15 | 141.98±10.11 |
| PT | 60.57±10.25 | 6.17±1.12 | 138.98±10.54 | 3.78±1.37 | 2.94±1.27 | 3.10±1.31 | 37.65±5.76 | 14.49±2.52 | 58.99±8.34 |
| SimVQ | 40.09±6.53 | 23.96±2.56 | 38.11±6.67 | 37.29±4.85 | 40.47±6.54 | 45.90±7.35 | 16.08±4.11 | 42.22±8.34 | 21.71±5.27 |
| HQA-GAE | 130.06±5.52 | 164.77±14.15 | 93.67±11.32 | 166.32±10.98 | 114.08±10.15 | 98.73±7.82 | 92.17±8.66 | 89.05±8.23 | 72.86±7.79 |
| RGVQ | **211.69**±5.27 | **319.09**±10.40 | **188.17**±11.23 | **446.02**±15.82 | **413.10**±10.78 | **228.82**±5.96 | **200.93**±7.89 | **374.51**±11.13 | **250.79**±8.63 |

*Table 2.* Cross-domain and cross-task performance in the pre-training and fine-tuning setting. Metrics are reported in terms of ROC-AUC for Graph Classification and Accuracy for all other tasks. **Bold** highlight the best performance.

| Method | Node Classification | | | Link Classification | | Graph Classification | | |
|---|---|---|---|---|---|---|---|---|
| | Cora | PubMed | WikiCS | WN18RR | FB15K237 | HIV | PCBA | *Avg.* |
| GCN | 75.65±1.37 | 75.61±2.10 | 75.28±1.34 | 73.79±0.39 | 82.22±0.28 | 64.84±4.78 | 71.32±0.49 | 74.10 |
| GAT | 76.24±1.62 | 74.86±1.87 | 76.28±0.78 | 80.16±0.27 | 88.93±0.15 | 65.54±6.93 | 70.12±0.89 | 76.01 |
| GIN | 73.59±2.10 | 69.51±6.87 | 49.77±4.72 | 74.02±0.55 | 83.21±0.53 | 66.86±3.48 | 72.69±0.22 | 69.95 |
| DGI | 72.10±0.34 | 73.13±0.64 | 75.32±0.95 | 75.75±0.59 | 81.34±0.15 | 59.62±1.21 | 63.31±0.89 | 71.51 |
| BGRL | 71.20±0.30 | 75.29±1.33 | 76.53±0.69 | 75.44±0.30 | 80.66±0.29 | 63.95±1.06 | 67.09±1.00 | 72.88 |
| GraphMAE | 73.10±0.40 | 74.32±0.33 | 72.61±0.39 | 78.99±0.48 | 85.30±0.16 | 61.04±0.55 | 63.30±0.78 | 72.66 |
| GIANT | 75.13±0.49 | 72.31±0.53 | 76.56±0.88 | 84.36±0.30 | 87.45±0.54 | 65.44±1.39 | 61.49±0.99 | 74.68 |
| GFT | 78.35±1.07 | 73.39±1.68 | 79.13±0.32 | 90.87±0.25 | 89.89±0.27 | 72.16±1.69 | 72.74±1.23 | 79.50 |
| GFT + EMA | 79.44±0.89 | 74.01±1.57 | 78.94±0.41 | 90.58±0.43 | 89.75±0.19 | 72.39±1.52 | 73.04±1.01 | 79.73 |
| GFT + AP | 79.69±1.07 | 75.05±0.86 | 79.73±0.35 | 89.56±0.18 | 89.05±0.18 | 71.86±1.53 | 71.48±0.99 | 79.48 |
| GFT + Reset | 80.07±0.91 | 75.51±0.69 | 79.85±0.33 | 91.18±0.43 | 88.09±0.23 | 72.79±1.65 | 71.95±0.85 | 79.92 |
| GFT + PT | 78.57±0.86 | 74.12±1.05 | 72.75±1.72 | 88.63±0.15 | 88.45±0.17 | 71.01±1.74 | 73.73±1.12 | 78.18 |
| GFT + SimVQ | 77.61±0.73 | 76.41±1.28 | 76.57±0.68 | 82.72±0.53 | 82.03±0.35 | 66.57±1.35 | 69.90±0.91 | 75.97 |
| GFT + RGVQ | **80.85**±0.73 | **77.46**±0.94 | **80.10**±0.52 | **91.32**±0.26 | **90.45**±0.31 | **74.10**±1.49 | **75.68**±0.99 | **81.42** |

and report perplexity in Table 1. Across all datasets, RGVQ outperforms all baselines by a clear margin and remains robust, showing that Gumbel-Softmax reparameterization combined with structure-aware regularization leads to more balanced codebook use, prevents collapse, and enables more expressive graph representations. By contrast, vanilla Graph VQ and its variants suffer from severe codebook collapse, with perplexity values as low as 1.00 in several datasets. More advanced mitigation strategies like SimVQ and HQA-GAE offer only small improvements.

**Transferability**. To evaluate the effectiveness of RGVQ in learning transferrable graph tokens, we integrate it into a graph foundation model, i.e., GFT, and compare it with vanilla Graph VQ and its variants with SOTA mitigation strategies. Moreover, we include supervised GNNs, i.e., GCN (Zhang et al., 2022), GAT (Veličković et al., 2018), and GIN (Xu et al., 2019), and graph self-supervised methods, i.e., DGI (Veličković et al., 2019), BGRL (Thakoor et al., 2022), GraphMAE (Hou et al., 2022), and GI-ANT (Chien et al., 2022). The supervised GNNs are trained directly on target dataset, while the self-supervised methods and all GFT variants are pretrained on the full set of datasets

and then fine-tuned per target. Table 2 reports cross-domain and cross-task performance. RGVQ consistently achieves the best performance across all tasks and datasets, surpassing both supervised and self-supervised models. It not only ensures better codebook utilization but also yields consistent downstream improvements.

**Serialization**. To further evaluate the effectiveness of RGVQ in serialization, we integrate it into the Graph Quantized Transformer (GQT), where discrete tokens serve as the input sequence to a vanilla Transformer backbone. We follow the original sequence reconstruction method (Wang et al., 2025) and compare the performance of RGVQ-enhanced GQT against GQT with different anti-collapse methods, supervised GNNs, and graph transformers, including GraphGPS (Rampášek et al., 2022), SGFormer (Wu et al., 2023), Exphomer (Shirzad et al., 2023), and Node-Former(Wu et al., 2022). Table 3 summarizes the node classification results across various benchmarks. Compared to GQT with conventional anti-collapse methods, incorporating RGVQ consistently improves classification accuracy on most datasets. While gains on some datasets are modest, this is expected, as discrete tokens serve as intermediate rep-

*Table 3.* Mean performance on node classification tasks. Metrics are reported in terms of ROC-AUC for Questions, and Accuracy for all other datasets. **Bold** indicates the best performance.

| | Cora | PubMed | Citeseer | Photo | Computer | WikiCS | Ratings | Roman | Questions |
|---|---|---|---|---|---|---|---|---|---|
| GCN | 75.65±1.37 | 78.80±0.60 | 71.60±0.40 | 92.70±0.20 | 89.65±0.52 | 77.47±0.85 | 48.70±0.63 | 73.69±0.74 | 76.09±1.27 |
| GAT | 76.24±1.62 | 79.00±0.40 | 72.10±1.10 | 93.87±0.11 | 90.78±0.13 | 76.91±0.82 | 52.70±0.62 | 88.75±0.41 | 76.79±0.71 |
| GraphGPS | 82.84±1.03 | 79.94±0.26 | 72.73±1.23 | 95.06±0.13 | 91.19±0.54 | 78.66±0.49 | 53.10±0.42 | 82.00±0.61 | 71.73±1.47 |
| SGFormer | 84.50±0.80 | 80.30±0.60 | 72.60±0.20 | 95.10±0.47 | 91.99±0.76 | 73.46±0.56 | 48.01±0.49 | 79.10±0.32 | 72.15±1.31 |
| Exphomer | 82.77±1.38 | 79.46±0.35 | 71.63±1.19 | 95.35±0.22 | 91.47±0.17 | 78.54±0.49 | 53.51±0.46 | 89.03±0.37 | - |
| NodeFormer | 83.20±0.90 | 79.90±1.00 | 72.50±1.10 | 93.46±0.35 | 86.98±0.62 | 74.73±0.94 | 43.86±0.35 | 64.49±0.73 | 74.27±1.46 |
| GQT | 86.44±1.58 | 81.60±1.35 | 73.14±1.26 | 94.46±0.68 | 92.13±0.23 | 80.03±0.19 | 54.04±0.12 | 89.85±0.73 | 76.52±1.52 |
| GQT + EMA | 86.23±1.19 | 81.41±1.24 | 73.08±1.58 | 94.01±0.57 | 91.95±0.18 | 79.98±0.23 | 54.10±0.08 | 89.91±0.51 | 75.94±1.16 |
| GQT + AP | 85.89±0.94 | 83.31±0.97 | 72.56±1.38 | 96.15±1.21 | 94.46±0.36 | 82.03±0.59 | 54.54±0.24 | 90.46±0.52 | 76.96±1.17 |
| GQT + Reset | 86.15±1.07 | 83.50±1.01 | 71.59±1.37 | 95.15±0.55 | 94.79±0.48 | 82.84±0.23 | 54.41±0.17 | 90.50±0.42 | 78.13±0.98 |
| GQT + PT | 85.71±1.44 | 80.92±1.15 | 79.53±1.23 | 94.74±0.76 | 92.35±0.35 | 75.65±0.78 | 54.50±0.14 | 89.76±0.68 | 76.74±1.34 |
| GQT + SimVQ | 86.02±1.64 | 82.56±1.02 | 72.58±1.14 | 95.21±0.77 | 94.23±0.21 | 81.78±0.32 | 53.98±0.15 | 90.15±0.66 | 76.35±1.21 |
| GQT + RGVQ | **88.34**±1.32 | **86.54**±1.41 | **81.25**±1.01 | **97.66**±1.05 | **95.67**±0.36 | **83.58**±0.66 | **55.16**±0.19 | **90.98**±0.66 | **78.26**±1.07 |

*Table 4.* Drop-one ablation.

| Variant | Cora | | PubMed | | WikiCS | |
|---|---|---|---|---|---|---|
| | Perp. | Acc. | Perp. | Acc. | Perp. | Acc. |
| Variant-1 | 94.47 | 78.35 | 4.14 | 73.39 | 10.18 | 79.13 |
| Variant-2 | 172.32 | 79.87 | 215.35 | 76.32 | 153.35 | 79.84 |
| Variant-3 | 135.45 | 79.12 | 208.16 | 76.29 | 179.49 | 79.79 |
| RGVQ | **211.69** | **80.85** | **319.09** | **77.46** | **228.82** | **80.10** |

resentations. In this setting, the backbone architecture (e.g., GQT's Residual VQ) and supervision during fine-tuning largely determine the final performance and can compensate for imperfections in tokenization.

## 6.2. Ablation Studies

We further conduct ablation experiments on the contributions of the Gumbel–Softmax reparameterization and the structure-aware contrastive regularization. Then, we vary the key hyperparameters in each module to evaluate their individual contributions. In addition, we provide additional ablation studies, convergence analysis, and an investigation of the effect of the number of GNN layers in Appendix B.

**Drop-one Ablation**. As the structure-aware regularization relies on the soft assignment probabilities produced by the Gumbel–Softmax reparameterization, without reparameterization, one-hot assignments provide no gradient to inactive codewords, making the regularization ineffective. Conversely, using soft assignments without regularization does not constrain the assignment distribution and therefore makes RGVQ behave similarly to vanilla VQ. Therefore, we remove either structural signals or feature signals to construct contrastive sets and assess three variants: removing all proposed components (variant-1), reparameterization with only structural regularization (variant-2), and reparameterization with only feature regularization (variant-3). As shown in Table 4, excluding either structural signals or fea-

ture signals reduces quantization diversity and consequently harms downstream accuracy. Removing Gumbel-softmax causes RGVQ to degenerate to vanilla VQ and leads to severe collapse. These results indicate that Gumbel-Softmax reparameterization and structure-aware regularization are mutually dependent in preventing codebook collapse, and both topological and feature information are essential for enhancing quantization diversity.

**Influence of the Temperature**. We investigate how the Gumbel–Softmax temperature $\tau$ affects perplexity. As shown in Figure 4a, lower temperatures, which produce distributions closer to one-hot, consistently improve codebook utilization. This indicates that, unlike some prior work (Zeng et al., 2025) that rely on temperature annealing, a relatively low and fixed temperature is sufficient to regularize the codebook distributions and address the non-differentiability of deterministic VQ.

**Influence of Contrastive Samples**. We examine how the number of contrastive samples in structure-aware regularization affects perplexity. As shown in Figure 4b, even a small number of contrastive samples (e.g., $n = 10$ or $50$) achieves relatively high perplexity, indicating effective codebook utilization. Increasing $n$ does not consistently lead to better utilization and may even degrade in some datasets, potentially due to added training noise.

**Perplexity and Downstream Performance**. To understand the relations between token diversity and the downstream performance, we evaluate how perplexity affects the node classification results. We select different pretraining checkpoints of RGVQ to reflect different perplexities. At the downstream stage, we utilize the pretrained tokens and fine-tune with node labels. Based on Figure 4c, we observe a consistent positive correlation between perplexity and accuracy. This suggests that a more diverse codebook can capture finer-grained structural patterns, enabling the model to learn more discriminative embeddings. It should be noted

*Table 5.* Ablation performance of variants under varying levels of heterophily.

| Variant | PubMed (0.19) | | | | Ratings (0.61) | | | | Roman (0.95) | | | |
|---|---|---|---|---|---|---|---|---|---|---|---|---|
| | Perp. | Acc. | Feat. Recon. | Link Recon. | Perp. | Acc. | Feat. Recon. | Link Recon. | Perp. | Acc. | Feat. Recon. | Link Recon. |
| Feat. only | 290.45 | 75.85 | **3.45e-4** | 9.14e-1 | 194.41 | 51.91 | **2.07e-3** | 8.90e-1 | **379.33** | **88.20** | **3.54e-3** | 9.06e-1 |
| Adj. only | 245.46 | 76.42 | 3.06e-3 | **6.59e-1** | 163.31 | 51.72 | 5.98e-3 | **6.69e-1** | 284.48 | 83.62 | 1.27e-2 | **6.80e-1** |
| Feat. & Adj. | 154.14 | 74.13 | 3.79e-4 | 7.31e-1 | 89.96 | 47.08 | 2.91e-3 | 7.73e-1 | 181.60 | 83.08 | 4.83e-3 | 7.54e-1 |
| **RGVQ** | **319.09** | **77.46** | 3.52e-4 | 6.93e-1 | **200.93** | **52.83** | 2.10e-3 | 6.93e-1 | 374.51 | 87.49 | 3.65e-3 | 6.93e-1 |

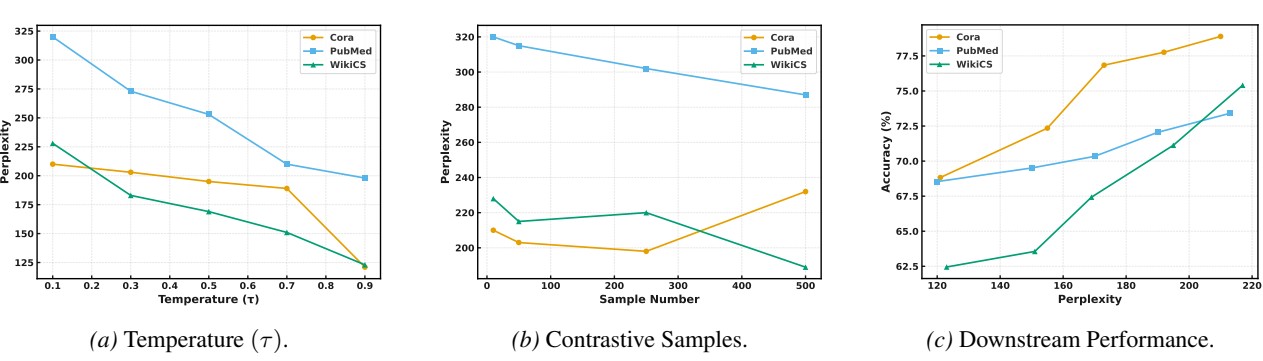

(a) Temperature ($\tau$).    (b) Contrastive Samples.    (c) Downstream Performance.

*Figure 4.* Ablation study with varying parameters.

that while the optimal codebook size may vary depending on the trade-off between compression and expressiveness, our model provides a flexible framework that maintains token diversity and scales to larger codebook sizes.

**Influence of Heterophily**. To evaluate the effectiveness of contrastive set design under varying levels of heterophily, we compare four variants for positive sets: feature-similar samples only (Feat. only), adjacent-only (Adj. only), their intersection (Feat. & Adj.), and RGVQ, while keeping the negative sets unchanged as defined in Equation 14. Table 5 summarizes quantization results, node classification and graph reconstruction performance on graph datasets with varying heterophily ratios. First, with respect to perplexity and downstream performance, feature-only positive sets achieve comparable perplexity to RGVQ, as they provide sufficient positives to regularize assignments and prevent collapse. However, lacking structural constraints, neighboring nodes may be quantized into different tokens, which can impair downstream performance in graphs with homophily. In highly heterophilous graphs, such as Roman-empire, feature-only positives slightly outperform RGVQ, since adjacent-only positives may introduce semantic noise. For adjacent-only and intersection-based variants, the absence of feature-level constraints weakens semantic consistency, and the limited number of positive samples causes InfoNCE to be dominated by negatives, leading to unstable optimization and degraded performance. Regarding reconstruction performance, RGVQ achieves the best balance between feature and link reconstruction. Omitting either feature-similar or adjacency positives results in a clear trade-off, indicating that single-type positives cannot fully preserve graph information.

## 7. Conclusion

In this paper, we investigate the codebook collapse in Graph VQ. Through empirical studies, we show that codebook collapse is not incidental, but a systematic issue in graphs. We diagnose underlying causes and propose RGVQ, a differentiable method that integrates both graph topology and feature similarity as explicit regularization to enhance codebook utilization. Extensive experiments demonstrate that RGVQ significantly mitigates collapse and improves downstream performance, highlighting its applicability in learning expressive and transferable graph representations.

## Impact Statement

**Ethical Aspects**. To the best of our knowledge, this work does not raise any specific ethical concerns. All datasets used in our experiments are publicly available benchmark datasets and have been widely adopted in prior research. Our method focuses on improving representation learning techniques and does not involve sensitive attributes.

**Societal Consequences**. This work studies a fundamental methodological issue in graph machine learning, namely codebook collapse in discrete graph tokenization, with the goal of improving model robustness and representation quality. The contributions of this paper are empirical studies of vector quantization on graphs and previous methods, the diagnosis of codebook collapse, and the proposed solution. We do not identify any immediate or specific societal consequences that warrant special discussion here.

## Acknowledgements

Xiaoyang Wang is supported by ARC DP240101322 and DP260100689. Wenjie Zhang is supported by Australian Research Council Centre of Excellence for Mathematical Modelling of Cellular Systems CE230100001 and Australian Research Council Discovery Project DP260100689.

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

*Table 6.* Dataset statistics for selected datasets.

| Dataset | Domain | Task | # Graphs | Avg. #Nodes | Avg. #Edges | # Classes | Metric |
|---|---|---|---|---|---|---|---|
| CiteSeer | Citation | Node | 1 | 3,327 | 4,522 | 6 | Accuracy |
| Cora | Citation | Node | 1 | 2,708 | 10,556 | 7 | Accuracy |
| PubMed | Citation | Node | 1 | 19,717 | 88,651 | 3 | Accuracy |
| Computer | Co-purchase | Node | 1 | 13,752 | 491,722 | 10 | Accuracy |
| Photo | Co-purchase | Node | 1 | 7,650 | 238,163 | 8 | Accuracy |
| WikiCS | Web link | Node | 1 | 11,701 | 216,123 | 10 | Accuracy |
| Amazon-Ratings | Review | Node | 1 | 22,662 | 32,927 | 18 | Accuracy |
| Roman-Empire | Synthetic | Node | 1 | 24,492 | 93,050 | 5 | Accuracy |
| Questions | Synthetic | Node | 1 | 48,921 | 153,540 | 2 | ROC-AUC |
| FB15K237 | Knowledge | Link | 1 | 14,541 | 310,116 | 237 | Accuracy |
| WN18RR | Knowledge | Link | 1 | 40,943 | 93,003 | 11 | Accuracy |
| PCBA | Molecule | Graph | 437,929 | 26.0 | 28.1 | 128 | ROC-AUC |
| HIV | Molecule | Graph | 41,127 | 25.5 | 27.5 | 2 | ROC-AUC |

## A. Experimental Setup

### A.1. Dataset

We use both homophilous and heterophilous graphs in our experiments. To implement empirical study and evaluate codebook utilization, we use various datasets, including Cora (Bojchevski & Günnemann, 2018), CiteSeer, PubMed (Namata et al., 2012), Amazon-Computer, Amazon-Photo (Shchur et al., 2018; McAuley et al., 2015), WikiCS (Mialon et al., 2021), Amazon-Ratings (Platonov et al., 2023), and Roman-Empire (Platonov et al., 2023). To assess transferability, we use cross-task and cross-domain datasets. Specifically, we use Cora, PubMed, and WikiCS for node classification; WN18RR (Shang et al., 2019) and FB15K237 (Li et al., 2024b) for link prediction; and HIV (Hu et al., 2022) and PCBA (Chen et al., 2024) for graph classification. Finally, we evaluate serialization ability using the same datasets employed for codebook utilization. Detailed dataset statistics are summarized in Table 6.

### A.2. Baseline

We use different baselines for the empirical study and three parts of our main experiments.

**Empirical Study and Codebook Utilization**. We primarily adopt codebook mitigation strategies originally developed in the vision and language domains, including EMA (Łańcucki et al., 2020), affine parameters (Huh et al., 2023), codebook reset (Zeghidour et al., 2021), and pretrained encoders (Zhao et al., 2024). We further include SimVQ (Zhu et al., 2025), which addresses the codebook collapse via one-single MLP layer over the latent basis vectors. Additionally, we compare with HQA-GAE (Zeng et al., 2025), a recent graph VQ model that applies a hierarchical VQ structure and an annealing strategy for codeword selection.

**Transferability**. To evaluate the effectiveness of RGVQ in learning transferrable graph tokens, we integrate it into a graph foundation model, i.e., GFT, and compare it with vanilla Graph VQ and its variants with different mitigation strategies. Moreover, we include supervised GNNs, i.e., GCN, GAT, and GIN, and graph self-supervised methods, i.e., DGI (Veličković et al., 2019), BGRL (Thakoor et al., 2022), GraphMAE (Hou et al., 2022), and GAINT (Chien et al., 2022). The supervised GNNs are trained directly on each target dataset, while the self-supervised methods and all GFT variants are pretrained on the full set of datasets and then fine-tuned per target.

**Serialization**. To further evaluate the effectiveness of RGVQ in serialization, we integrate it into the Graph Quantized Transformer (GQT) (Wang et al., 2025), where discrete tokens serve as the input sequence to a vanilla Transformer backbone. We follow the original sequence reconstruction method and compare the performance of RGVQ-enhanced GQT against the original GQT, supervised GNNs, and graph transformers, including GraphGPS (Rampášek et al., 2022), SGFormer (Wu et al., 2023), Exphomer (Shirzad et al., 2023), and Nodeformer (Wu et al., 2022).

*Table 7.* Hyperparameters of RGVQ for each dataset.

| Hyperparameter | Cora | Pubmed | Citeseer | Computer | Photo | WikiCS | Ratings | Roman | Questions |
|---|---|---|---|---|---|---|---|---|---|
| Hidden dimension | 256 | 256 | 256 | 256 | 256 | 256 | 256 | 256 | 256 |
| Learning rate | 0.001 | 0.001 | 0.001 | 0.001 | 0.001 | 0.001 | 0.001 | 0.001 | 0.001 |
| Weight decay | 1e-5 | 1e-5 | 1e-5 | 1e-5 | 1e-5 | 1e-5 | 1e-5 | 1e-5 | 1e-5 |
| Seed | 42 | 42 | 42 | 42 | 42 | 42 | 42 | 42 | 42 |
| Epochs | 1000 | 1000 | 1000 | 1000 | 1000 | 1000 | 1000 | 1000 | 1000 |
| Feature Reconstruction | 100 | 100 | 100 | 100 | 100 | 100 | 100 | 100 | 100 |
| Topology Reconstruction | 0.01 | 0.01 | 0.01 | 0.01 | 0.01 | 0.01 | 0.01 | 0.01 | 0.01 |
| $\beta$ | 1 | 1 | 1 | 1 | 1 | 1 | 1 | 1 | 1 |
| Temperature | 0.1 | 0.1 | 0.1 | 0.1 | 0.1 | 0.1 | 0.1 | 0.1 | 0.1 |
| Similarity function | Cosine | Cosine | Cosine | Cosine | Cosine | Cosine | Cosine | Cosine | Cosine |
| Top-$K$ | 20 | 20 | 20 | 20 | 20 | 20 | 20 | 20 | 20 |
| Sample number | 50 | 50 | 50 | 50 | 50 | 50 | 50 | 50 | 50 |
| GNN layers | 4 | 4 | 4 | 4 | 4 | 4 | 4 | 4 | 4 |

## A.3. Implementation Details

**Empirical Study**. We provide the hyperparameters and experimental setup used in the empirical study of codebook perplexity. We jointly train the single-head VQ model and the GAT encoder using the link prediction and feature reconstruction tasks, along with the commitment loss and vocabulary loss. The task weights are set to 0.01, 100, 0.1, and 0.9, respectively. We train the model for 1000 epochs and report the highest perplexity during the training process for each method and a specific codebook size $K$. For all methods, we utilize the kmeans initialization and orthogonal regulation for the codebook, with a regularization weight of 0.1. The GNN consists of 4 layers with a hidden dimension of 256. AdamW is utilized as the optimizer with a learning rate of 1e-4 and a weight decay of 1e-5. For affine parameters, we use Euclidean distance and set the codebook decay to 0.9. For codebook reset, the threshold of deadcode is set to 10. For pretraining encoder, we pretrain the GNN encoder for 50 epochs before the joint training.

**Codebook Utilization**. The implementation of all collapse mitigation strategies is the same with the empirical study. For HQA-GAE, we use one-head VQ model and use the same hidden dimension and number of GNN layers as other methods, while all remaining hyperparameters follow the original paper. For RGVQ, we set the codebook size $K$ to 512. To construct the contrastive sample sets, for each node, we construct a pool of positive candidates by combining its 1-hop neighbors with the nodes that are most 20 similar in the input feature space (top-$K = 20$), which is measured by cosine similarity. From this pool, we sample 50 nodes with replacement as positive samples. The negative pool is defined symmetrically as all nodes that are neither neighbors nor feature positives, and we sample 50 negative nodes with replacement from the negative pool. The training weights for the link reconstruction, node feature reconstruction, contrastive regularization, commitment loss, and vocabulary loss are set to 0.01, 100, 1, 0.1, and 0.9, respectively. We set the temperature for the Gumbel-Softmax trick to 0.1. We train on each dataset for 1000 epochs to ensure convergence, and repeat the process 20 times to report the mean perplexity with standard deviations. Detailed hyperparameters for each dataset are summarized in Table 7.

**Transferability**. We use RGVQ as a plugin within the pretraining pipeline of GFT. Specifically, we retain the same pretraining tasks in GFT (Wang et al., 2024), including the link, node feature, and node embedding reconstruction tasks, and integrate RGVQ as a regularization term. Their weights are set to 100, 1, 0.01, and 10, respectively. For the backbone encoder, we utilize a 2-layer GCN model with ReLU activation, and set the codebook size to 512 and the hidden dimension to 256. We use AdamW optimizer with a learning rate of 1e-3 and weight decay of 1e-5. For data augmentation, we apply a link drop rate and the node-feature drop rate of 0.2. We pretrain the VQ tokens for 500 epochs on all datasets. During finetuning, we repeat each experiment 20 times to report the average performance with standard deviations. We finetune the model for 250 epochs using early stopping. For dataset splits, we follow the commonly used protocol for Cora and PubMed and utilize the predefined 10 splits with different seeds to report the downstream performance. Each split includes 20 labeled nodes per class for training. For WikiCS, we follow the recommended protocol by OGB and use the official split, reporting average performance across 20 splits (Mernyei & Cangea, 2020). For WN18RR, we utilize 86,835/3,034/3,134 links for training/validation/test, respectively. For FB15K237, we use 272,155/17,535/20,466 links for

*Table 8.* Selected hyperparameters in GQT for each dataset.

| Dataset | GNN Encoder | | Quantizer | | Transformer | | | | |
|---|---|---|---|---|---|---|---|---|---|
| | # layers | # Hidden dim | # Codebooks | Codebook size | KNN | PPR | # Layers | # Heads | # FFN dim |
| Cora | 2 | 256 | 3 | 128 | 0 | 15 | 2 | 4 | 512 |
| CiteSeer | 2 | 256 | 3 | 128 | 5 | 15 | 2 | 4 | 512 |
| PubMed | 2 | 256 | 3 | 256 | 0 | 15 | 2 | 4 | 512 |
| Computer | 2 | 256 | 3 | 128 | 5 | 30 | 2 | 4 | 512 |
| Photo | 3 | 512 | 3 | 128 | 5 | 20 | 2 | 4 | 1024 |
| WikiCS | 2 | 256 | 3 | 128 | 5 | 30 | 2 | 4 | 512 |
| Amazon-Ratings | 4 | 512 | 3 | 128 | 5 | 20 | 2 | 4 | 1024 |
| Roman-Empire | 6 | 256 | 3 | 256 | 10 | 15 | 3 | 4 | 512 |
| Questions | 3 | 256 | 3 | 512 | 10 | 15 | 2 | 4 | 512 |

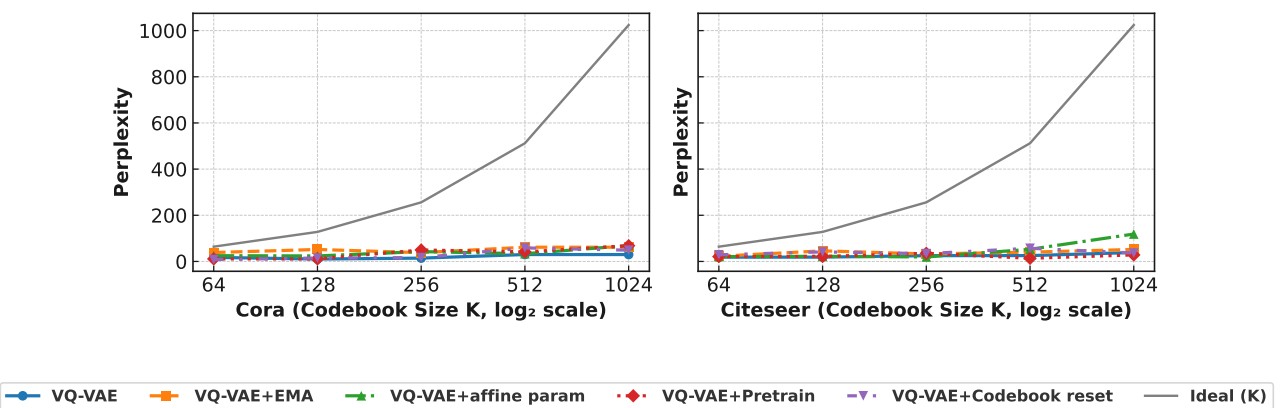

*Figure 5.* Codebook perplexites on graph datasets. The black lines indicate the optimal perplexities, i.e., codebook size K.

training/validation/test, respectively. For HIV and PCBA, we follow the official data split and utilize 80%/10%/10% for training/validation/test set (Hu et al., 2020).

**Serialization**. The training of GQT includes two parts: the VQ tokenizer and the backbone transformer. We detail the implementations and training hyperparameters below. For the VQ tokenizer, we follow the original paper and use Residual VQ (Wang et al., 2025). We retain all of the reconstruction tasks in the pretraining setting of GQT, including Deep Graph Infomax (DGI) (Veličković et al., 2019) and GraphMAE2 (Hou et al., 2023), and integrate RGVQ as a regularization term. For the tokenizer, we set the number of codebooks to three for GQT, GQT + EMA, GQT + AP, GQT + Reset, GQT + PT; and one for GQT + SimVQ, GQT + RGVQ. We choose codebook size from {128,256,512}. For the GNN encoder, we adopt GCN with ReLU activation, varying the number of layers from {2,3,4,6} and hidden dimensions from {256,512}. We pretrain the VQ tokenizer and the GNN encoder for 200 epochs until convergence. For training the vanilla transformer, we construct semantic links using K-Nearest-Neighbors, with K in {0,5,10}. To serialize the input graph sequence, we use Personalized PageRank (PPR) to generate a sequence for each node, with the sequence length selected from {15,20,30}. The transformer uses 2 or 3 layers, 4 attention heads, and a feedforward dimension of {512, 1024}. The detailed hyperparameters are summarized in Table 8. We train transformers with node labels together with the pretrained VQ tokenizer and GNN encoder, and report the average performance and standard deviations over 5 runs. For Cora, Pubmed, Citeseer, Computer, Photo, we follow the original settings in (Wang et al., 2025), using 60%/20%/20% for training/validation/test. For WikiCS, we follow the predefined split in (Mernyei & Cangea, 2020) and report the average performance across 20 splits. For Amazon-Ratings, Roman-Empire, and Questions, we adopt the splits in (Platonov et al., 2023), using 50%/25%/25% for training/validation/test, and report the mean performance over 10 random splits.

## B. Additional Experiments

**Empirical Study**. We additionally provide the quantization results on Cora and Citeseer datasets. The results shown in Figure 5 suggest that codebook collapse is a systematic problem in Cora and Citeseer datasets, even though the mitigation

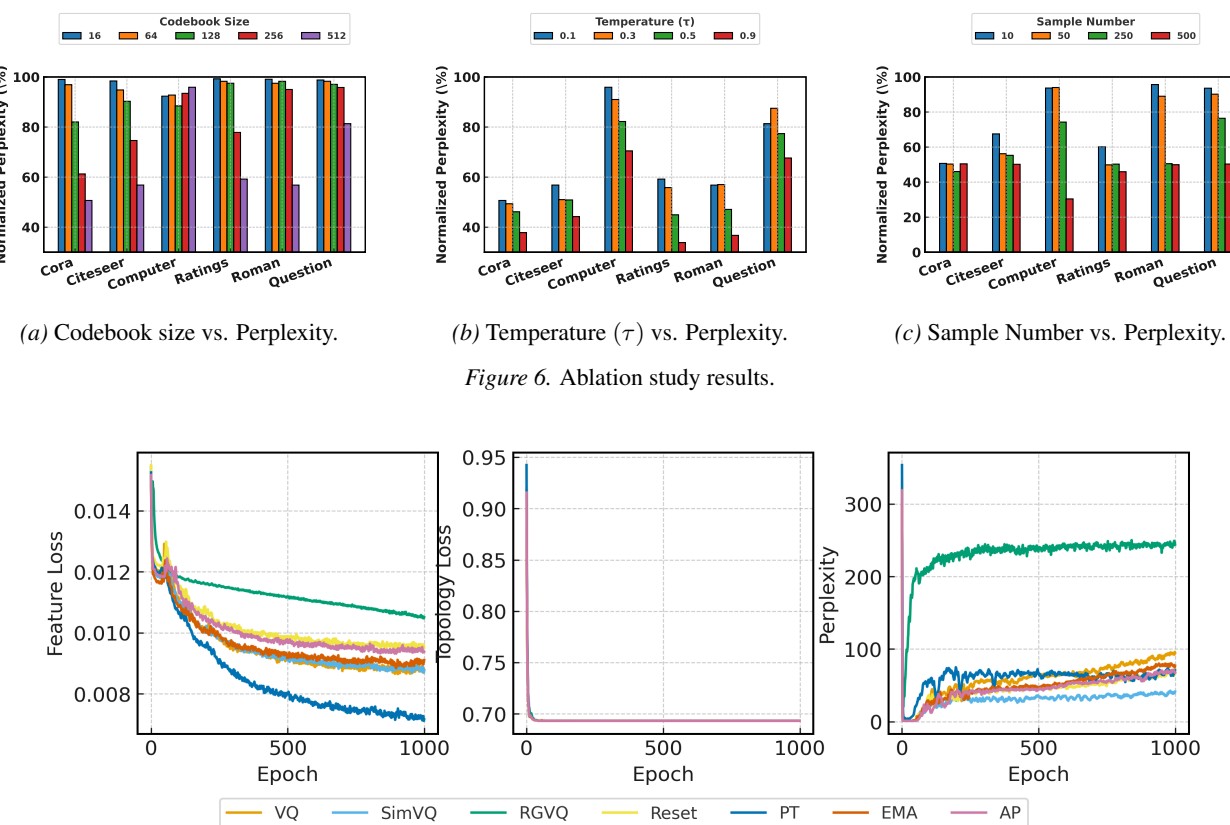

*(a)* Codebook size vs. Perplexity.      *(b)* Temperature ($\tau$) vs. Perplexity.      *(c)* Sample Number vs. Perplexity.

*Figure 6.* Ablation study results.

*Figure 7.* Reconstruction loss and perplexity during the pretraining process on Cora

strategies are applied.

**Codebook diversity on real datasets**. We use PCA@95 as a proxy for effective feature redundancy and the average node degree as a proxy for local connectivity. We provide statistics on eight real graphs in Table 9, from which we observe that datasets with higher average degree (Photo, Computer) or lower PCA@95 (Ratings, Roman, Questions) exhibit lower perplexity, whereas datasets with both higher PCA@95 and lower degree (Cora, Citeseer) exhibit much weaker collapse. Additionally, we perform a multivariate regression where perplexity is modeled as a function of PCA@95 and Avg. degree. We report regression coefficients ($\beta$), Pearson coefficients, partial correlations between each variable and perplexity, and their p-values in Table 10. PCA@95 shows a positive correlation, while Avg. degree exhibits a negative trend but does not reach statistical significance. These results are consistent with the trends observed in Figure 2. Because the analysis is based on limited datasets, which may constrain the statistical power, we treat this as empirical evidence rather than a causal conclusion.

**Ablation Study**. We also provide the additional ablation study to further evaluate the contribution of each proposed module in RGVQ in this section. Here we use the normalized perplexity, defined as the ratio of utilized codebook entries to the total size of the codebook. First, we evaluate how codebook size affects the codebook utilization of RGVQ. The results are shown in Figure 6a. Across all datasets, RGVQ consistently maintains high normalized perplexity as the codebook size increases, showing strong robustness to the choice of codebook size. Notably, even with a large codebook size of 512, the model utilizes over 50% of the codebook capacity across all datasets. Second, we investigate how the Gumbel–Softmax temperature $\tau$ affects normalized perplexity. A lower temperature yields a codebook assignment distribution that is closer to one-hot. As shown in Figure 6b, unlike some prior work (Zeng et al., 2025) that rely on temperature annealing, a relatively low and fixed temperature is sufficient to address the non-differentiability of deterministic VQ on selected datasets. Finally, we examine how the number of contrastive samples in structure-aware regularization affects normalized perplexity. As shown in Figure 6c, even a small number of contrastive samples (e.g., $n = 10$ or $50$) achieves relatively high normalized perplexity, indicating effective codebook use. Increasing $n$ does not consistently lead to better utilization and may even

*Table 9.* Dataset statistics (PCA@95, average degree) and measured codebook perplexity across 8 graph datasets.

| Dataset | PCA@95 | Avg Degree | Perplexity |
|---------|--------|------------|------------|
| Cora | 802 | 4.90 | 94.47 |
| Pubmed | 410 | 5.50 | 4.14 |
| Citeseer | 1459 | 3.74 | 60.09 |
| Photo | 611 | 32.13 | 1.00 |
| Computer | 646 | 36.76 | 1.00 |
| Ratings | 194 | 8.60 | 13.29 |
| Roman | 141 | 3.91 | 10.84 |
| Questions | 160 | 7.28 | 20.78 |

*Table 10.* Multivariate regression of PCA@95, average degree, and perplexity.

| | PCA@95 | Avg Degree |
|---|--------|------------|
| Pearson $r$ (p-value) | 0.714 (0.013) | -0.249 (0.461) |
| Partial $r$ (p-value) | 0.733 (0.010) | -0.336 (0.312) |
| $\beta$ (p-value) | 0.026 (0.016) | -0.230 (0.342) |

degrade in some datasets, potentially due to added training noise.

**Converge Analysis**. We also provide the reconstruction loss and perplexity curves during the pretraining process of RGVQ and all baselines with codebook size $K = 512$. As shown in Figure 7, all baselines reach stable reconstruction loss within the first 250 epochs and remain stable afterwards, while they all collapse to less than 100 and do not recover. This confirms that the collapse is a problem for vanilla VQ and other anti-collapse solutions. Regarding reconstruction performance, collapse does not necessarily produce large reconstruction losses because a strong decoder can overfit to node features or links even though usable tokens are limited. However, this phenomenon is fundamentally undesirable. When nodes collapse to the small portion of tokens, the discrete latent space becomes degenerate and ceases to reflect any structural or semantic diversity in the graph. In this situation, the VQ module fails to provide meaningful discrete representations. While the appropriate codebook size may depend on task-specific trade-offs between compression and expressiveness, our method offers a flexible and effective framework that preserves token diversity while scaling to larger codebook sizes.

**Influence of GNN Layer Number**. To better understand the relations between GNN layer number and quantization diversity, we evaluate how the number of GNN layer $L$ affects the quantization perplexity in RGVQ and vanilla VQ. Based on the results in Table 11, we make the following observations: (1) As the layer number $L$ increases, the perplexity of vanilla VQ consistently decreases. Deeper GNNs suffer from over-smoothing, causing node representations to fall more easily within the radius of the same codeword, resulting in less diverse quantization. (2) RGVQ is robust for different layers because it provides explicit regularization.

## C. Complexity Analysis

Assume a $L$-layer GNN, a codebook of size $K$, and hidden dimension of $d$, the number of nodes and links are denoted as $|\mathcal{V}|$ and $|\mathcal{E}|$ respectively. We divide the complexity analysis into two parts: Pre-computation of contrastive set and quantization process.

**Pre-computation of Contrastive Set**. Before training, RGVQ constructs for each node sets of positive and negative samples, based on both structural and feature similarity. This step is performed once and reused during training. To implement this, neighbors are first extracted by scanning the adjacency matrix, which requires $O(|\mathcal{E}|)$ time. Then compute feature distances between each node and all others will take $O(|\mathcal{V}|^2 d)$ time. However, in practice, we adopt a sampling strategy: for each node, we sample $M$ non-neighbor nodes (where $M$ is a small constant, e.g., 100) and compute their feature similarity. This limits the total cost of semantic similarity computation to $O(|\mathcal{V}| \cdot M \cdot d)$, which is linear in the number of nodes. After collecting both structurally and semantically similar candidates, we perform top-$k$ selection for each node to finalize its positive sample set, costing $O(|\mathcal{V}| \log k)$ time in total. Negative pairs are sampled from the set of all nodes excluding the positives. Thus, the overall time complexity of the contrastive set construction process is $O(|\mathcal{E}| + |\mathcal{V}| \cdot M \cdot d + |\mathcal{V}| \log k)$,

*Table 11.* Perplexity with varying number of GNN layers $L$.

| Dataset | $L=1$ | | $L=2$ | | $L=3$ | | $L=4$ | | $L=5$ | |
| --- | --- | --- | --- | --- | --- | --- | --- | --- | --- | --- |
| | VQ | RGVQ | VQ | RGVQ | VQ | RGVQ | VQ | RGVQ | VQ | RGVQ |
| Cora | 154.34 | 394.96 | 121.59 | 339.19 | 109.06 | 257.47 | 94.47 | 211.69 | 99.44 | 218.45 |
| Pubmed | 8.97 | 452.16 | 3.12 | 300.51 | 5.18 | 295.64 | 4.14 | 319.09 | 4.07 | 295.64 |
| Photo | 1.99 | 432.32 | 1.00 | 421.04 | 1.00 | 443.60 | 1.00 | 446.02 | 1.00 | 306.06 |
| Computer | 3.81 | 468.98 | 1.00 | 452.41 | 1.00 | 464.65 | 1.00 | 413.10 | 1.00 | 394.83 |
| Ratings | 32.64 | 414.10 | 15.59 | 295.28 | 10.80 | 213.42 | 13.29 | 200.93 | 9.14 | 207.18 |

*Table 12.* Performance on large graphs.

| Dataset | VQ | EMA | AP | Reset | PT | SimVQ | RGVQ |
| --- | --- | --- | --- | --- | --- | --- | --- |
| Arxiv | $6.43 \pm 0.94$ | $7.42 \pm 1.13$ | $52.39 \pm 5.56$ | $121.24 \pm 9.46$ | $7.85 \pm 1.05$ | $54.34 \pm 7.03$ | $\mathbf{305.13} \pm 20.30$ |
| Products | $10.13 \pm 1.05$ | $13.45 \pm 1.79$ | $67.71 \pm 7.84$ | $105.23 \pm 15.45$ | $8.79 \pm 0.98$ | $61.12 \pm 13.23$ | $\mathbf{223.47} \pm 10.81$ |
| Proteins | $9.85 \pm 0.86$ | $14.87 \pm 1.42$ | $49.32 \pm 7.51$ | $123.74 \pm 20.01$ | $8.47 \pm 0.82$ | $56.63 \pm 9.72$ | $\mathbf{254.87} \pm 10.67$ |

which is linear in the number of nodes and edges under fixed $M$ and $k$.

**Quantization Process**. The time and space complexity of the GNN encoder are $O(Ld^2|\mathcal{V}| + Ld|\mathcal{E}|)$ and $O(Ld^2 + Ld|\mathcal{V}| + |\mathcal{E}|)$, respectively. The decoder has the same complexity. RGVQ computes the distance between each node embedding and all $K$ codewords, and estimates the soft assignment distribution via the Gumbel-Softmax trick. This process requires $O(|\mathcal{V}|Kd)$ time and $O(|\mathcal{V}|K)$ space. Finally, RGVQ regularizes the assignment distributions using the InfoNCE loss between node pairs. For each node, this involves computing similarities with $k$ positive and $k$ negative samples, each over $K$-dimensional distributions. The total cost is $O(|\mathcal{V}| \cdot (2k) \cdot K)$.

**Scalability on Large Graphs**. The model is easy to scale to large graphs, as the same strategy can be adopted for each minibatch. For example, we implement the same sampling strategy and construct the contrastive set in each input mini-batch. We additionally include the performance of RGVQ against vanilla VQ on ogbn-arxiv, ogbn-product, and ogbn-proteins in Table 12.

