# OpenReview forum: "Graph is a Natural Regularization: Revisiting Vector Quantization for Graph Representation Learning"
_ICML.cc/2026/Conference — ICML 2026 regular_

### Official Review · Reviewer_TyMo · 2026-02-26

**Soundness:** 3
**Presentation:** 2
**Significance:** 2
**Originality:** 2
**Overall Recommendation:** 4
**Confidence:** 4

**Summary:**

This paper examines codebook collapse in Graph Vector Quantization, a phenomenon where only a subset of codewords receives meaningful updates during training. The authors focus on investigating a key problem regarding the relationship between graph structural properties, such as feature redundancy and connectivity density, and the severity of collapse. Building on their diagnosis, this paper address this issue by proposing RGVQ, which combines Gumbel-Softmax reparameterization for differentiable assignment with a structure-aware contrastive regularization term. Experimental evaluation covers codebook utilization, transferability within a graph foundation model framework, and serialization compatibility with transformer-based architectures.

**Compliance With Llm Reviewing Policy:**

Affirmed.

**Final Justification:**

I have no further questions. I suggest that the authors provide a more detailed discussion in future versions on how their method differs from approaches such as HQA-GAE. In addition, some theoretical analysis would make the paper more convincing.

**Key Questions For Authors:**

Please see the above weakness.

**Strengths And Weaknesses:**

**Strengths**

S1. The correlation analysis between graph-level statistics (PCA@95%, average degree) and perplexity provides a useful empirical lens for understanding why collapse may be more pronounced in graph domains compared to vision or language.

S2. This paper is easy to follow.

S3. The experiments are comprehensive.

**Weaknesses**

W1. The claim of providing "the first empirical study" of codebook collapse on graphs appears overstated. Prior work such as HQA-GAE  explicitly addresses codebook underutilization and the "winner-takes-all" phenomenon in graph VQ, proposing hierarchical quantization and annealing-based selection as mitigation. The current submission does not adequately differentiate RGVQ from HQA-GAE in terms of mechanistic advantages or empirical gains specifically attributable to utilization improvement.

W2. Section 4.2 offers an intuitive narrative about a "self-reinforcing feedback loop" induced by hard assignment. While plausible, this analysis remains at the level of heuristic explanation. The connection between graph properties and collapse is empirically suggestive but lacks causal or theoretical grounding.

W3. RGVQ combines two well-established techniques: Gumbel-Softmax for relaxing discrete sampling and InfoNCE-style contrastive regularization on representation distributions. Applying these to VQ assignment distributions is a reasonable adaptation, but the core algorithmic novelty is limited.

W4. There is noticeable redundancy in the manuscript that affects readability. For instance, the statement regarding the empirical observation of collapse in the Introduction (lines 64-77: "From the data perspective...") is nearly identical to the text in the Motivation section (lines 143-152: "From a data perspective...").

---

> ### Author Rebuttal · Authors · 2026-03-30
>
> We thank the reviewer for the insightful and constructive feedback. We address each concern in detail below.
>
> **W1 (1): Comparison with HQA-GAE**
>
> First, we acknowledge that both our work and HQA-GAE focus on codebook underutilization. Our method aims to mitigate collapse by breaking the dynamics of hard assignment in a way that accounts for graph data characteristics. Specifically, RGVQ leverages soft assignments over all codewords (Eq. 12) and incorporates structure-aware regularization. This enables task signals to control the quantization process and gradients to propagate to all codewords. Combined with Gumbel noise, it further reduces the tendency toward winner-takes-all behavior.
>
> In contrast, HQA-GAE mitigates codebook underutilization through hierarchical quantization and annealing-based selection. While effective, it primarily relies on multi-level codebooks and indirect updates. As the temperature decreases and assignments become increasingly deterministic, rarely selected codewords may still receive limited task-driven gradients. In addition, its hierarchical design updates underutilized codewords mainly through cluster-level aggregation, which may not fully reflect task signals.
>
> Empirically, we include HQA-GAE as a baseline in Table 1, where RGVQ consistently outperforms it across datasets, demonstrating improved codebook utilization and downstream performance. We will further clarify this comparison in the revised version.
>
> **W1 (2): "First Empirical Study" Claim:**
>
> We respectfully clarify that our “first empirical study” claim is not intended to suggest that prior work has not observed underutilization. Instead, we refer to a systematic empirical analysis of codebook collapse in graph VQ, including comparisons across widely used mitigation strategies (e.g., EMA, pretraining, codebook reset, and affine parameters). In addition, our study provides detailed diagnostics (e.g., codebook size scaling behavior) and analysis from both data (e.g., feature redundancy, connectivity) and optimization (e.g., hard assignment dynamics) perspectives, which have been less explored in prior work. We agree that the original statement may be interpreted as too broad and will revise it to more precisely reflect this contribution.
>
> **W2: Graph Properties and Collapse:**
>
> We clarify that our goal in Section 4.2 is not to establish a formal causal theory, but to provide an empirical diagnosis grounded in optimization dynamics.
>
> To provide empirical support for the connection between graph properties and collapse, we conduct regression analysis by modeling perplexity as a function of graph properties (PCA@95 and Avg Degree).
>
> | Metric                     | PCA@95           | Avg Degree        |
> |--------------------------|------------------|-------------------|
> | Pearson r (p-value)      | **0.714 (0.013)** | -0.249 (0.461)    |
> | Partial r (p-value)      | **0.733 (0.010)** | -0.336 (0.312)    |
> | Regression β (p-value)   | **0.026 (0.016)** | -0.230 (0.342)    |
>
> PCA@95 shows a positive effect, and average degree yields negative coefficients, although it does not reach statistical significance. This is consistent with the trends observed in Figure 2 and the diagnosis that graph redundancy contributes to collapse. We emphasize that these findings are correlational and serve as empirical evidence rather than a causal explanation. We will revise the manuscript to clarify this interpretation.
>
> **W3: Limited Novelty:**
>
> We acknowledge that our paper does not introduce entirely new primitives. Our contribution lies in a problem-driven framework that combines empirical study, diagnosis, and solution design for graph VQ: we systematically analyze codebook collapse, identify the role of graph redundancy and hard assignment dynamics, and develop an assignment-level regularization to mitigate it.
>
> First, we provide a systematic empirical study of widely used VQ techniques on graphs, revealing their limitations in preventing codebook collapse. In particular, we identify the correlation between graph redundancy and collapse, and highlight the role of hard assignment dynamics in the training.
>
> Based on this analysis, RGVQ combines (1) assignment-level optimization via soft assignments that enable gradient flow to all codewords, and (2) structure-aware regularization that incorporates graph topology and feature similarity into the assignment process. Different from previous contrastive learning, our method operates on token assignment distributions rather than embedding space, and directly regulates codebook utilization.
>
> **W4: Redundancy in Writing:**
>
> We thank the reviewer for pointing this out. We will further improve the writing for better readability.

---

> > ### Author Rebuttal · Reviewer_TyMo · 2026-04-03
> >
> > Thanks for the detailed response and experiments. I will raise my score.

---

### Official Review · Reviewer_vhUY · 2026-02-27

**Soundness:** 2
**Presentation:** 2
**Significance:** 2
**Originality:** 2
**Overall Recommendation:** 3
**Confidence:** 4

**Summary:**

This paper presents a framework called RGVQ to mitigate the codebook collapse issue in Graph Vector Quantization. The manuscript diagnoses its root causes from data redundancy and optimization dynamics, and combines Gumbel-Softmax soft assignments with structure-aware regularization to penalize concentrated token usage. Experiments show that augmenting existing backbones with RGVQ enhances codebook utilization, transferability, and serialization.

**Compliance With Llm Reviewing Policy:**

Affirmed.

**Key Questions For Authors:**

1.Could the authors clarify why the hyperparameter values in Table 6 are almost identical? Although it is located in the Appendix, it remains unclear whether these results reflect actual findings or a potential typographical error?

2.Could the authors provide an empirical comparison of the training time and memory footprint per epoch between RGVQ and efficient baselines? How much practical overhead does the structure-aware regularization introduce?

3.In Appendix C, the paper mentions sampling M non-neighbor nodes to approximate the Top-K feature-similar nodes. Could the authors provide an empirical trade-off curve between training time and performance as M varies?

**Limitations:**

1.The regularization term proposed in the paper exhibits a homophily bias, which may pose limitations in heterophilous scenarios. Currently, the manuscript lacks an in-depth discussion regarding this potential constraint.

2.The authors should further clarify the detrimental impact of codebook collapse, either through additional empirical evidence or by incorporating relevant citations.

3.The manuscript lacks an empirical analysis of the computational complexity associated with the proposed constraint term.

**Strengths And Weaknesses:**

Strengths

1.This paper analyzes the problem from the perspectives of data and optimization, proposing that graph redundancy and non-i.i.d. characteristics exacerbate collapse.

2.RGVQ integrates Gumbel-Softmax with contrastive regularization, elegantly translating graph structural inductive biases into constraints that avert collapse.

3.This paper verifies on canonical tasks and investigate the plug-and-play potential across different graph architectures, alongside a rigorous validation of scalability on large-scale datasets.

Weakness
1.In constructing the positive sample set (Eq. 13), RGVQ takes the union of 1-hop neighbors and top-K feature-similar nodes. While reasonable for homophilous graphs, enforcing similar token distributions across topological neighbors in heterophilous scenarios warrants careful consideration.

2.I recommend that the authors enhance Figure 3; in its current form, the illustration is overly simplistic and obscures the core contributions of the work.

3.The authors should explicitly clarify the detrimental effects of codebook collapse. The  description at line 55—stating that 'a limited number of tokens can be utilized... leading to overly coarse representations'—is somewhat vague. Providing a more precise mechanism for this degradation, along with relevant citations, would significantly strengthen the argument.

---

> ### Author Rebuttal · Authors · 2026-03-30
>
> We thank the reviewer for all the feedback.
>
> **Q1 - Hyperparameters for RGVQ:**
>
> We confirm that this is not a typo. We performed hyperparameter tuning across datasets and observed that performance is insensitive to these hyperparameters within a reasonable range, with similar configurations consistently achieving near-optimal results. Therefore, we report a unified configuration to highlight robustness. We acknowledge that Table 6 may appear redundant and will clarify this in the revision.
>
> **Q2, L3 - Computation Overhead:**
>
> We utilize sampling (size $M$) to avoid full graph similarity computation and select $Top-k$ contrastive samples from sampled sets to compute InfoNCE. The additional computation of RGVQ comes from similarity and contrastive computation, with $O(|\mathcal{V}| M d)$ and $O(|\mathcal{V}| 2k d)$. Both introduce low overhead compared to GNN encoding $O(Ld^2|\mathcal{V}| + Ld|\mathcal{E}|)$. We use full-graph training for small graphs (sampling precomputed before training) and mini-batch training for large graphs (sampling computed per batch, Line 967). We evaluate on PubMed (full graph, 19,717 nodes) and OGBN-Products (mini-batch=512, 2.5M nodes), reporting runtime and GPU memory. We set $M=50$ and $Top-k = 10$.
>
> |          | Time per Epoch (PubMed) | GPU Memo (PubMed) | Time per Batch Step (Products) | GPU Memo (Products) |
> |--------------------------|------------------|------------------------|------------------------------|------------------------|
> | RGVQ                     | 0.35 s           | 1600 MB               | 0.25 s                       | 900 MB                |
> | VQ / VQ with EMA and Reset | 0.30 s         | 1400 MB               | 0.20 s                       | 800 MB                |
> | SimVQ                    | 0.35 s           | 1570 MB               | 0.25 s                       | 870 MB                |
> | AP                       | 0.45 s           | 1550 MB               | 0.35 s                       | 900 MB                |
> | HQA-GAE                  | 0.80 s           | 2400 MB               | 0.50 s                       | 1500 MB               |
>
> RGVQ improves quantization with ~30% additional runtime and ~10% more memory over VQ.
>
> **Q3 - Computation Overhead of Sampling:**
> As discussed above, from $M$ candidates, Top-k nodes are selected as contrastive samples. Since we already examined the influence of Top-K in Figure 4 (b), we fix $k$ as 20 and report the computation overhead of GFT+RGVQ on PubMed  with varying $M$:
>
> |                | $M = 20$ | $M = 50$ | $M = 100$ | $M = 1000$ |
> |----------------|--------|--------|---------|----------|
> | Perplexity     | 315.28 | 319.09 | 320.33  | 329.36   |
> | Construction Time | 0.042 s | 0.053 s | 0.063 s  | 0.069 s  |
> | Training Time  | 0.351 s | 0.357 s | 0.364 s  | 0.372 s  |
> | Acc.            | 77.59  | 77.46  | 77.45   | 77.33    |
>
> Increasing $M$ has a negligible impact on performance but slightly increases computation. Overall, RGVQ achieves robust performance with acceptable overhead.
>
> **L1, W1 - Positive Samples for Heterophily:**
> We acknowledge that incorporating adjacent nodes as positives may introduce a homophily bias, which could be a potential limitation in heterophilous graphs. However, we clarify why this design is appropriate.
>
> First, VQ operates in the embedding space rather than label semantics. Due to GNN message passing, adjacent nodes share largely overlapping computation graphs and correlated embeddings. Distance-based VQ operates on these embeddings and naturally tends to assign similar tokens to adjacent nodes. Encouraging similar assignment distributions is therefore consistent with such a mechanism. Second, this design aligns with the link reconstruction objective in VQ training, which reconstructs adjacency rather than enforcing semantic separation. Setting neighbors as positives generally has fewer reconstruction losses.
>
> Finally, we use the union of adjacent and feature-similar nodes as positives because restricting positives to nodes that are both adjacent and semantically similar would significantly reduce positive samples, especially in mini-batch settings.
>
> **W3, L2 - Impacts of Collapse:**
>
> Codebook collapse corresponds to extremely low perplexity, meaning only a small subset of codes is utilized under a desired granularity. As shown in Fig. 4(c) in the paper, lower perplexity consistently leads to degraded representation quality and worse downstream accuracy. We further include Photo dataset, where severe collapse occurs. During RGVQ pretraining, perplexity gradually increases and stabilizes. We select checkpoints with different perplexities and evaluate downstream performance, confirming that lower perplexity generally yields worse results.
> We will include relevant citations to clarify this impact.
>
> | Perplexity | 32.93 | 64.86 | 147.01 | 250.72 | 396.63 |
> |------------|-------|-------|--------|--------|--------|
> | Acc        | 78.65 | 82.36 | 90.77  | 94.01  | 96.64  |

---

> > ### Author Rebuttal · Reviewer_vhUY · 2026-04-03
> >
> > I thank the authors for the detailed rebuttal addressing my questions. I am currently weighing whether to revise my score.

---

> > > ### Author Response · Authors · 2026-04-07
> > >
> > > Dear reviewer,
> > >
> > > Thank you for your positive feedback. We are glad that our clarifications addressed your concerns regarding efficiency,  heterophily, and the impacts of collapse. We will definitely incorporate these explanations into the final version to further improve clarity and completeness. We also hope these additions better highlight the effectiveness and robustness of our approach.

---

### Official Review · Reviewer_mKEQ · 2026-03-08

**Soundness:** 3
**Presentation:** 3
**Significance:** 3
**Originality:** 3
**Overall Recommendation:** 4
**Confidence:** 4

**Summary:**

This paper studies codebook collapse in vector quantization for graph data. It first shows empirically that collapse occurs consistently across multiple graph datasets, and that mitigation strategies from vision and language only provide limited improvement in the graph setting. The paper then analyzes the issue from the perspectives of graph data properties and optimization dynamics. Based on this, it proposes RGVQ, which combines Gumbel-Softmax soft assignment with structure-aware contrastive regularization to reduce the self-reinforcing collapse behavior and redundant token co-assignment. Experiments on codebook utilization, transferability, and serialization show consistent gains over existing methods.

Overall, I find the paper well motivated and the problem important. The method is practical and the empirical study is fairly broad. That said, I still have concerns about the fairness of part of the experimental comparison, the strength of some empirical claims, and the validity of certain design assumptions in heterophilous settings. For these reasons, I currently lean Weak Reject, though I would be open to raising my score if the key issues are addressed convincingly.

**Compliance With Llm Reviewing Policy:**

Affirmed.

**Final Justification:**

Most of my major concerns have now been addressed. In particular, the added same-codebook comparison improves the fairness of the serialization comparison, the additional statistical analysis makes the correlation claim more convincing, and the new targeted ablation on heterophilous graphs provides the direct empirical support that I previously felt was missing. These additions substantially strengthen the paper.

I am still not fully satisfied with the limitation discussion, which remains somewhat brief and could more clearly articulate failure cases or settings where the proposed positive-set design may be less suitable. That said, I no longer think this issue outweighs the paper’s overall merits.

**Key Questions For Authors:**

In Table 3, RGVQ uses one codebook while the other GQT baselines mostly use three. Could the authors provide results under the same number of codebooks? This would make the serialization comparison much more convincing.

The positive set \mathcal{N}_P includes adjacent nodes, but in heterophilous graphs adjacent nodes often differ semantically. How do the authors justify this design, and why does it still help in heterophilous settings?

For Figure 2, could the authors report Spearman or Pearson correlation coefficients, together with p-values? This would help support the correlation claim more rigorously.

**Limitations:**

The paper does not discuss failure scenarios for RGVQ: under what graph characteristics or downstream task settings the method may fail to yield meaningful performance gains, or where the positive set construction assumption may produce negative effects. It also does not address whether the computational overhead remains acceptable on very large graphs with tens of millions of nodes.

**Strengths And Weaknesses:**

Strengths:
The paper targets a meaningful and underexplored problem. Codebook collapse is a real bottleneck for graph VQ methods, especially if such tokenization schemes are to support graph foundation models.

The empirical study is reasonably comprehensive. The experiments cover both homophilous and heterophilous graphs, as well as transfer and serialization settings. The additional large-graph results and convergence analysis in the appendix also strengthen the empirical case.

The method is practical. RGVQ can be integrated into both GFT and GQT as a plug-in module, and the ablation study shows that the two proposed components work together in a complementary way.

Weaknesses:
The serialization comparison in Table 3 is not fully controlled. According to Appendix A.3, RGVQ and GQT+SimVQ use one codebook, while the other GQT baselines use three. Since the number of codebooks affects the modeling setup and capacity, this makes the comparison difficult to interpret cleanly. This difference is also not made explicit in the main text, which weakens the credibility of the serialization results.

The correlation claim in Figure 2 is based on limited evidence. The analysis uses only eight datasets, and no correlation coefficients or significance tests are reported. The observed trend is interesting, but the current presentation is stronger than the evidence seems to support.

The paper does not sufficiently discuss the positive-set construction in heterophilous graphs. The regularization includes adjacent nodes as positives, but in heterophilous settings adjacent nodes are often semantically dissimilar. The paper does not clearly explain why this design remains appropriate in such cases.

The paper would benefit from a clearer explanation of how the token-distribution-level contrastive regularization differs from more standard contrastive objectives applied at the node representation level, both in motivation and in effect.

---

> ### Author Rebuttal · Authors · 2026-03-29
>
> We thank the reviewer for raising these important concerns.
>
> **W1, Q1- Codebook Fairness:**
>
> We follow the original GQT paper, which adopts a 3-layer Residual VQ (each layer is a standard VQ), where each consecutive codebook quantizes the residual error from the previous one. A 1-layer setting reduces Residual VQ and its variants (e.g., EMA Residual VQ) to vanilla VQ. Therefore, we keep 3 codebooks for GQT and its variants, while GQT+RGVQ uses one codebook, as RGVQ does not rely on residual layers. For fair comparisons, we additionally include results of GQT with one codebook
>
> | | GQT  | EMA  | AP   | Reset | PT   | RGVQ |
> |---------|------|------|------|-------|------|------|
> | WikiCS  | 78.41 | 78.46 | 80.19 | 81.03 | 74.79 | **83.58** |
> | Ratings | 52.37 | 52.65 | 53.31 | 53.12 | 53.58 | **55.16** |
> | PubMed  | 79.01 | 78.96 | 83.91 | 83.24 | 78.81 | **86.54** |
> | Roman   | 87.38 | 87.25 | 88.06 | 88.34 | 87.16 | **90.98** |
>
> RGVQ consistently outperforms GQT baselines under the same number of codebooks by enhancing quantization diversity without depending on multiple codebooks. We will clarify in the main text.
>
> **W2, Q3 - Correlation of Collapse and Graph Redundancy:**
>
> We perform a multivariate regression where perplexity is modeled as a function of PCA@95 and Avg. degree. We report regression coefficients (β) and p-values, and additionally compute Pearson and partial correlations between each variable and perplexity:
>
> |   | PCA@95           | Avg Degree        |
> |--------------------------|------------------|-------------------|
> | Pearson r (p-value)      | 0.714 (0.013) | -0.249 (0.461)    |
> | Partial r (p-value)      | 0.733 (0.010) | -0.336 (0.312)    |
> | β (p-value)   | 0.026 (0.016) | -0.230 (0.342)    |
>
> PCA@95 shows a positive correlation, while Avg. degree exhibits a negative trend but does not reach statistical significance. These results are consistent with the trends in Figure 2. Because the analysis is based on limited datasets, which may constrain the statistical power, we treat this as empirical evidence rather than a causal conclusion.
>
> **W3 and Q2 - Positive Sets for Heterophily:**
>
> We clarify the reasonableness of including adjacent nodes as positives and why this is effective for downstream tasks for heterophilous graphs.
>
> First, the positive set is defined in the embedding space rather than label semantics. Due to the message passing of general GNNs adopted in our paper, adjacent nodes have largely overlapping computation graphs, resulting in correlated embeddings. As distance-based VQ operates on these embeddings, it naturally tends to assign similar tokens to adjacent nodes. Encouraging similar assignment distributions for adjacent nodes is therefore consistent with the quantization mechanism. Notably, similar assignment distributions do not imply same quantization outputs, especially with the stochasticity introduced by Gumbel noise $g_{ik}$.
> Second, this design is aligned with the VQ link reconstruction objective, where the goal is to preserve adjacency information rather than enforce semantic separation. Encouraging adjacent nodes to have consistent assignments generally has fewer reconstruction losses.
>
> Although requiring positive samples to be both adjacent and feature-similar seems to be efficient to distinguish node semantics, it would make positive pairs extremely limited for some nodes, especially in mini-batch training for large graphs, leading to dominance of negative samples. Thus, we use either adjacent or feature-similar nodes as positives.
>
> Empirically, RGVQ remains effective on heterophilous graphs when integrated into the VQ-based backbone (GQT in Table 3). By mitigating codebook collapse, RGVQ consistently improves performance over GQT with other VQ variants. This suggests that the proposed positive-set design does not hinder learning in heterophilous settings.
>
> **W4: Regularization on Distribution Level:**
> We find that applying contrastive objectives in the embedding space (either node or quantized embeddings) cannot address self-reinforced loops, as rarely selected codewords still receive little gradients.
> In contrast, RGVQ leverages soft assignments over all codewords (Eq. 12). By regularizing the assignment over all codewords, it directly influences the quantization process and ensures that gradients can propagate to all codewords through soft assignments. Combined with Gumbel noise, it breaks the self-reinforced loops.
>
> **L1: Computation Overhead on Large Graphs:**
> We utilize sampling (size $M$) to avoid full graph similarity computation on large graphs and select $Top-k$ contrastive samples for InfoNCE. The additional costs come from similarity and InfoNCE computation, with $O(|\mathcal{V}| M d)$ and $O(|\mathcal{V}| 2k d)$, resulting in ~30% additional runtime and ~10% more memory over VQ variants. Under $M=50$  and $Top-k=10$, RGVQ incurs only marginal overhead compared to VQ, with 0.25 s vs. 0.20 s per batch step and 900 MB vs. 800 MB GPU memory.

---

> > ### Author Rebuttal · Reviewer_mKEQ · 2026-04-03
> >
> > Thank you for the detailed rebuttal. The authors addressed several of my earlier concerns constructively. In particular, the added statistical analysis strengthens the correlation claim, and the clarification of the proposed distribution-level regularization improves the methodological clarity. I also appreciate the clarification regarding the codebook-setting issue in the serialization experiments.
> >
> > That said, I still view two points as only partially resolved.
> >
> > First, for the positive-set construction in heterophilous graphs, the rebuttal provides a clearer methodological justification, but this point is still supported more by mechanism-level reasoning than by direct empirical validation. A small targeted ablation on heterophilous datasets comparing adjacent-only, feature-similar-only, and union-based positive sets would make this much more convincing.
> >
> > Second, regarding limitations, the added discussion of computational overhead is helpful, but my original concern was broader than efficiency alone. The paper still does not sufficiently discuss failure scenarios, such as graph settings where the method may yield limited gains or where the positive-set assumption may be less appropriate. Even a brief analysis or discussion of such cases would strengthen this part.
> >
> > Overall, the rebuttal improves the paper, but these issues remain only partially resolved.

---

> > > ### Author Response · Authors · 2026-04-04
> > >
> > > We sincerely thank the reviewer for these constructive suggestions.
> > >
> > > **Concern1 - Ablation Study:**
> > >
> > > We use graph datasets under different heterophily ratios, ranging from homophilous to highly heterophilous settings (PubMed: 0.19, Amazon-Ratings: 0.61, Roman Empire: 0.95). We use feature-similar only (Variant-1), adjacent-only (Variant-2), intersection-based (Variant-3), and union-based (RGVQ) positive sets, while negative sets are maintained the same with Eq.14. We set codebook size as 512 and report converged perplexity, downstream node classification performance, and reconstruction losses.
> > >
> > > | Variant | PubMed (0.19) |  |  |  | Ratings (0.61) |  |  |  | Roman (0.95) |  |  |  |
> > > |---------|---------------|--|--|--|----------------|--|--|--|---------------|--|--|--|
> > > |         | **Perp.** | **Acc.** | **Feat. Recon.** | **Link Recon.** | **Perp.** | **Acc.** | **Feat. Recon.** | **Link Recon.** | **Perp.** | **Acc.** | **Feat. Recon.** | **Link Recon.** |
> > > | Variant-1 | 290.45 | 75.85 | **3.45e-4** | 9.14e-1 | 194.41 | 51.91 | **2.07e-3** | 8.90e-1 | **379.33** | **88.20** | **3.54e-3** | 9.06e-1 |
> > > | Variant-2 | 245.46 | 76.42 | 3.06e-3 | **6.59e-1** | 163.31 | 51.72 | 5.98e-3 | **6.69e-1** | 284.48 | 83.62 | 1.27e-2 | **6.80e-1** |
> > > | Variant-3 | 154.14 | 74.13 | 3.79e-4 | 7.31e-1 | 89.96 | 47.08 | 2.91e-3 | 7.73e-1 | 181.60 | 83.08 | 4.83e-3 | 7.54e-1 |
> > > | **RGVQ** | **319.09** | **77.46** | 3.52e-4 | 6.93e-1 | **200.93** | **52.83** | 2.10e-3 | 6.93e-1 | 374.51 | 87.49 | 3.65e-3 | 6.93e-1 |
> > >
> > > Based on the table, we make the following observations:
> > >
> > > **Ob1): Perplexity and downstream performance.** Feature-only positive sets achieve comparable perplexity to RGVQ, as both provide sufficient positives to regularize the assignment distribution and prevent severe collapse (Top-k = 10). However, feature-only signal constraints leave structurally adjacent nodes unconstrained. As a result, neighboring nodes are more likely to be quantized to different tokens, which may harm downstream performance, especially when homophilous signals are still present. In contrast, in extremely heterophilous graphs (e.g., Roman-empire), feature-only positive sets perform better, and enforcing adjacency-based positives may introduce semantic noise and lead to slight degradation.
> > >
> > > For adjacency-only variants, the lack of feature-based constraints weakens semantic alignment, which is harmful to downstream tasks. In addition, adjacency-only and intersection-based sets may provide limited positives, causing InfoNCE to be dominated by negatives and leading to unstable optimization and degraded performance.
> > >
> > > **Ob2): Reconstruction performance.** Our design of positive sets aligns with reconstruction objectives of VQ: preserving both node feature and graph structure. Accordingly, RGVQ achieves the best balance in both feature and link reconstruction. Removing either type in positive sets leads to a clear trade-off: feature-only variants degrade link reconstruction, while adjacency-only variants degrade feature reconstruction, indicating that single-type positive sets fail to preserve the full information of the graph.
> > >
> > > **Conclusion:** Although introducing adjacent nodes as positives may incur partial semantic bias in highly heterophilous settings, this does not significantly harm downstream performance. This is because InfoNCE operates on assignment distributions but not embedding-level, and similar assignment distributions do not necessarily imply identical quantization outputs, especially under the stochasticity introduced by Gumbel noise. Overall, combining both types of positives enables RGVQ to better preserve both feature and topological information.
> > >
> > > **Concern2 - Limitation Discussion:**
> > >
> > > While RGVQ is effective and robust across a range of graphs, its design of positive sets may introduce noisy semantic signals in cases of extreme heterophily. In such settings, feature-based positives may be more aligned with graph semantics. Moreover, current self-supervised objectives of RGVQ primarily focus on preserving topology and input features, and may not explicitly model higher-level semantic relations between nodes. As a result, in scenarios where semantic signals are inconsistent with graph topology, a more explicit semantic-aware reconstruction objective could be beneficial.
> > >
> > > We believe adding these analyses in the revised version will significantly improve our paper.

---

### Official Review · Reviewer_1XZS · 2026-03-22

**Soundness:** 3
**Presentation:** 2
**Significance:** 2
**Originality:** 2
**Overall Recommendation:** 4
**Confidence:** 4

**Summary:**

The paper studies the problem of codebook collapse in the context of Vector Quantization (VQ) applied to graphs. In particular, the authors show through experiments that when using discrete quantization techniques on node embeddings produced by GNNs, the codebook tends to be underutilized, meaning that nodes are mostly assigned to a small number of dominant tokens.

To analyze this phenomenon, the authors study both properties of graph data, such as feature redundancy and connectivity density, and the optimization dynamics of deterministic VQ, which lead to a positive feedback loop that favors a few codewords.

To address this issue, the authors propose a regularization framework called RGVQ, which combines a soft quantization via Gumbel-Softmax and a contrastive regularization based on graph structure and feature similarity.

The experiments show better codebook utilization (measured by perplexity) and improvements in downstream performance across several tasks.

**Compliance With Llm Reviewing Policy:**

Affirmed.

**Key Questions For Authors:**

- In Figure 1, the use of raw perplexity may be misleading when comparing different codebook sizes, since perplexity is upper-bounded by K. Have you considered reporting a normalized perplexity (e.g., P/K) in the main paper, or at least clarifying in the caption that the key point is the sub-linear growth with respect to K? A normalized metric could make the results easier to interpret (also in other tables).
- The method is integrated into GFT and GQT. Could you clarify why these specific frameworks were chosen? Is this due to architectural reasons or simply because they are available implementations?
- In the ablation study, on which types of datasets are the results reported? It would be useful to understand whether the behavior of the method changes between homophilous and heterophilous graphs.

**Limitations:**

The authors do not explicitly discuss the limitations of their work.

**Strengths And Weaknesses:**

_Strengths_
- The paper addresses a relevant problem in the emerging area of graph tokenization and graph foundation models.
- The paper is generally well written and well structured.
- The empirical analysis of codebook collapse is well supported by experiments.
- The proposed framework is partially convincing: the role of Gumbel-Softmax in solving the gradient issue is clear, while the regularization component is less convincing.
- The experiments are fairly complete and include ablation studies.


_Weaknesses_
- Some claims are supported mainly by correlations (e.g., the relation between perplexity and performance), which is acceptable but does not demonstrate causality.
- The use of raw perplexity as the main metric can be misleading when varying the codebook size K, since it is not normalized. This makes some results (e.g., Figure 1) harder to interpret correctly.
- The comparison with other methods does not appear fully fair. If I understood correctly, the proposed method benefits from multi-dataset pretraining, while other supervised baselines are not trained under the same conditions. This point should be clarified.
- Some methodological details are not clearly specified. For example, the similarity function used in the contrastive loss (Eq. 15) is not explicitly defined.
- In Table 1, it is not completely clear whether the separation between homophilous and heterophilous graphs is indicated by the vertical line. It would be helpful to clarify this and comment on any differences in performance between the two groups.
- In the main tables, it could be useful to highlight also the second-best result, to make performance gaps easier to interpret.
- In the Optimization Dynamics section, the authors state that in deterministic VQ only selected codewords receive updates, leading to a positive feedback loop where frequently selected codewords dominate while others remain inactive. If I understood correctly, this is the main mechanism behind collapse. It might be helpful to explain this idea more clearly and intuitively before introducing the formal equations.
- The analysis of graph properties is limited to PCA-based feature redundancy and average node degree, which may not fully capture the structural factors contributing to codebook collapse. It would be interesting to consider more expressive graph descriptors, such as centrality measures.
- In Figure 2, the authors discuss correlations between graph properties and perplexity, but it is not entirely clear from the text which datasets or settings are used to obtain these results. This should be clarified.

****

Typos / Minor Comments
- p. 6, line ~326: “and NodeFormer(Wu et al., 2022).”
- In Figure 6 (Ablation study), the y-axis

---

> ### Author Rebuttal · Authors · 2026-03-30
>
> We thank the reviewer for the constructive feedback.
>
> **Q1, W2 - Normalized Perplexity:**
>
> In our experiments, perplexity of VQ and variants cannot scale with increasing codebook size $K$, so normalized perplexity quickly collapses to near-zero values for most methods, making figures less visually informative. For this reason, we report raw perplexity and include the expected (ideal) scaling as a reference to highlight the collapse. To make it clear, we report the normalized perplexity on Citesser below:
>
> | K    | VQ | EMA   | AP | PT | Reset |
> |------|--------|-------|--------------|----------|----------------|
> | 64   | 0.29 | 0.35 | 0.32 | 0.32 | 0.45 |
> | 128  | 0.14 | 0.35 | 0.17 | 0.16 | 0.32 |
> | 256  | 0.09 | 0.12 | 0.07| 0.13 | 0.13 |
> | 512  | 0.05 | 0.07 | 0.10 | 0.02| 0.11 |
> | 1024 | 0.04 | 0.05 | 0.11 | 0.02 | 0.03 |
>
> We agree that reporting normalized perplexity can make Table 1 and Table 4 more interpretable, so we will report this in the revised version.
>
> **Q2- Why Choose GQT and GFT:**
>
> We choose GQT and GFT as backbones with clear motivations. As discussed in Lines 32–42, VQ serves two key roles in graph learning: (1) learning a reusable token vocabulary, and (2) enabling graph serialization into token sequences for Transformer-based models. Our backbone choices align with these two usages. GFT leverages VQ tokens as a transferable vocabulary for cross-task learning, and GQT uses VQ tokens as input sequences for Transformer models. Integrating our method into these frameworks allows us to directly evaluate whether RGVQ improves transferability and serialization capability in downstream tasks by addressing codebook collapse.
>
> **Q3 - Ablation Study:**
>
> For all ablation studies, we use homophilous graphs (Cora, PubMed, WikiCS). Here, we include two heterophilous graphs and report the classification results using direct VQ tokens. We will add them in the revised version.
>
> | Variant    | Ratings Perp. | Ratings Acc. | Roman Perp. | Roman Acc. |
> |------------|-----------:|----------:|-------------:|------------:|
> | Variant-1  | 13.29      | 45.74    | 10.84        | 73.19    |
> | Variant-2  | 145.54     | 50.13    | 207.69      | 83.27      |
> | Variant-3  | 153.63     | 50.08    | 253.65      | 83.53     |
> | **RGVQ**   | **200.93** | **52.83** | **374.51**   | **87.49**   |
>
> **W1 - Performance and Perplexity:**
>
> We agree that the observed relationship between perplexity and downstream performance is correlational and does not imply causality. We will revise the manuscript to avoid misunderstanding.  As shown in Fig. 4(c) in the paper, lower perplexity consistently leads to degraded representation quality and worse downstream accuracy. We further include Photo dataset, where severe collapse occurs. We select checkpoints with different perplexities and evaluate downstream performance, confirming that lower perplexity generally yields worse results.
>
> | Perplexity | 32.93 | 64.86 | 147.01 | 250.72 | 396.63 |
> |------------|-------|-------|--------|--------|--------|
> | Acc        | 78.65 | 82.36 | 90.77  | 94.01  | 96.64  |
>
> However, we do not claim that increasing perplexity alone is sufficient to improve performance, as downstream results also depend on backbone models and training dynamics.
>
>
> **W3 - Baseline Fairness:**
>
> For supervised GNNs, we follow the standard protocol and train directly on each target graph. Supervised training of GNNs across datasets is non-trivial due to the lack of shared label spaces across graphs, whereas VQ and other SSL methods are explicitly designed for transferable pretraining. Therefore, the comparison reflects the intended usage of each paradigm rather than an unfair comparison.
>
> **W4-7:**
>
> Same as Eq. 13, we adopt cosine similarity in Eq. 15. We will clarify these problems in the revised version for better readability.
>
> **W8: Other Graph Descriptors:**
>
> We select PCA@95 and average degree because they reflect feature and structural redundancies, respectively. We also analyse Katz centrality, which measures how important a node is based on how well it is connected to important nodes in the graph:
> | | Katz Centrality | Perplexity |
> |------------------|-------------|------------|
> | Cora             | 1.0258         | 94.47      |
> | Pubmed           | 1.0298         | 4.14       |
> | Citeseer         | 1.0193         | 60.09      |
> | Photo            | 1.3325         | 1.00       |
> | Computer         | 1.8263         | 1.00       |
> | Ratings          | 1.0459         | 13.29      |
> | Roman            | 1.0199         | 10.84      |
> | Questions        | 1.0704         | 20.78      |
>
> Katz centrality shows a weak negative association with perplexity: datasets with higher global connectivity tend to exhibit lower perplexity, although the trend is not strictly monotonic.
>
> **W9: Figure 2:**
>
> For Figure 2, we adopt the same setting in the empirical study in Section 4.1, with the implementation details in Appendix A. The detailed values and datasets can be found in Table 8.

---

> > ### Author Rebuttal · Reviewer_1XZS · 2026-04-03
> >
> > The rebuttal improves the work's positioning, but some explanations are not convincing, in particular the fairness of the comparisons.

---

> > > ### Author Response · Authors · 2026-04-04
> > >
> > > We thank the reviewer for the feedback and clarify our experimental design for cross-domain evaluation (Table 2).
> > >
> > > We include three paradigms: supervised GNNs, self-supervised GNNs, and VQ-based models (GFT). The key difference is that self-supervised and VQ-based methods are designed for transferable pretraining, as they rely on task-agnostic objectives, whereas supervised GNNs are inherently task-specific and depend on label supervision.
> > >
> > > **1 - Supervised GNNs.**
> > >
> > > We include supervised GNNs (e.g., GCN, GAT, GIN) as baselines, which rely on task-specific labels (node, link, or graph labels) for training. For this reason, we train supervised GNNs only on each target dataset. Extending them to cross-dataset training is non-trivial due to multiple sources of inconsistency:
> > > (i) **task-level mismatch:** our evaluation spans node, link, and graph classification, which require different prediction heads and loss functions, making it difficult to define a unified training objective across datasets;
> > > (ii) **label space inconsistency:** datasets differ in label semantics, preventing a shared output space or a reusable classifier.
> > >
> > > While one could introduce additional mechanisms (e.g., multi-head training or label alignment), such modifications are non-standard and would effectively change the learning paradigm. Therefore, we follow common settings of prior cross-domain works that include supervised GNNs as baselines and train supervised GNNs directly on each target dataset [1][2][3].
> > >
> > > **2 - Strong self-supervised baselines.**
> > >
> > > To ensure fairness within the pretraining and finetuning setting, we include multiple SOTA graph SSL methods (DGI, BGRL, GraphMAE, GIANT). These methods naturally support cross-dataset pretraining via task-agnostic objectives, and we follow their original protocols by pretraining on all datasets and finetuning on each target graph, consistent with VQ-based models.
> > >
> > > Overall, our comparisons respect the standard usage of each paradigm, while the transfer setting is strictly controlled among all pretraining-based methods, ensuring a fair and meaningful evaluation.
> > >
> > > We hope this clarification addresses the concern regarding the fairness of our comparisons.
> > >
> > > [1] GFT: Graph Foundation Model with Transferable Tree Vocabulary, NeurIPS 2024.
> > >
> > > [2] One for all: Towards training one graph model for all classification tasks. ICLR 2024.
> > >
> > > [3] All in One and One for All: A Simple yet Effective Method towards Cross-domain Graph Pretraining, KDD 2024.

---

### Decision · Program_Chairs · 2026-04-30

**Decision:**

Accept (regular)

**Comment:**

The paper investigates the problem of codebook collapse in graph vector quantization, where only a small subset of codewords is effectively utilized during training. The authors provide an empirical analysis, attributing the issue to graph data properties and to optimization dynamics in hard assignment. To address this issue, the authors propose RGVQ, which combines Gumbel-Softmax-based soft assignments with structure-aware contrastive regularization. Experimental results show improved codebook utilization and consistent gains across diverse scenarios.

Reviewers broadly agree on the following strengths of the paper.
- S1. The empirical analysis of codebook collapse is comprehensive and provides useful insights into the problem.
- S2. The proposed method is reasonably designed and has the practical advantage of being easily integrated into various backbones.
- S3. The experimental evaluation is comprehensive

Major concerns shared by the reviewers include
- W1. Some claims (e.g., correlation between perplexity and performance) lack a strong causal justification.
- W2. The overall level of novelty is viewed as somewhat limited.
- W3. Certain aspects of the method design remain questionable in heterophilous settings

During the rebuttal, W3 is largely addressed through additional analysis and experiments. W1 and W2 are partially mitigated through improved statistical testing and clarification, though not fully resolved. As a result, the overall assessment shifts slightly toward acceptance. The overall contributions outweigh the remaining limitations, and thus acceptance is recommended.